# FVEL: Interactive Formal Verification Environment with Large Language Models via Theorem Proving

**Xiaohan Lin**[1]* **Qingxing Cao**[1]* **Yinya Huang**[2]* **Haiming Wang**[3]* **Jianqiao Lu**[4]
**Zhengying Liu**[5] **Linqi Song**[2] **Xiaodan Liang**[1,6]†
[1]Shenzhen Campus of Sun Yat-sen University   [2]City Univeristy of Hong Kong
[3]Sun Yat-sen University   [4]The University of Hong Kong
[5]Huawei Noah's Ark Lab   [6]DarkMatter AI Research

## Abstract

Formal verification (FV) has witnessed growing significance with emerging program synthesis by the evolving large language models (LLMs). However, current formal verification mainly resorts to symbolic verifiers or hand-craft rules, resulting in limitations for extensive and flexible verification. On the other hand, formal systems for automated theorem proving, such as Isabelle, serve as another line of rigorous verification, upheld by extensive rules and theorems. In this paper, we propose FVEL[3], an interactive **F**ormal **V**erification **E**nvironment with **L**LMs. Specifically, FVEL transforms a given code to be verified into Isabelle, and then conducts verification via neural automated theorem proving with an LLM. The joined paradigm leverages the rigorous yet abundant formulated and organized rules in Isabelle and is also convenient for introducing and adjusting cutting-edge LLMs. To achieve this goal, we extract a large-scale dataset for automated formal verification named FVELER[3]. The FVELER dataset includes code dependencies and verification processes that are formulated in Isabelle, containing 758 theories, 29,304 lemmas, and 201,498 proof steps with in-depth dependencies. We benchmark FVELER in the FVEL environment by fine-tuning LLMs with FVELER and then evaluating them on Code2Inv and SV-COMP. The results show that FVEL with FVELER fine-tuned Llama3-8B solves 17.39% (69→81) more problems, and Mistral-7B 12% (75→84) more problems in SV-COMP. And the proportion of proof errors is reduced. Project page: https://fveler.github.io/.

## 1   Introduction

Formal verification (FV), or automated program verification [36, 2] checks if a code meets a specific demand and is correct to implement. As the code synthesis ability of current models [27, 24, 8] evolves rapidly, there is a growing demand for automated verification of diverse and abundant synthesis programs. However, current formal verification mainly resorts to symbolic verifiers [10, 6, 21] or hand-craft rules [40]. However, symbolic verification can not leverage the advanced reasoning ability of current large language models (LLMs), while hand-craft rules with limited execution on specific code cases have restricted abilities to general verification.

On the other hand, automated theorem proving (ATP) [43, 1, 13] is a line of work on rigorous verification with formal languages (e.g., Isabelle [29], Lean [5]) and interactive proof environments (e.g., PISA [15], LeanDojo [41]). Such formal languages and toolkits maintain corresponding

---

*   Equal contribution.
†   Corresponding author.
[3]FVEL: Pronounced as fuel. FVELER: Pronounced as fueler. FVEL **e**nvironment **r**esource.

38th Conference on Neural Information Processing Systems (NeurIPS 2024) Track on Datasets and Benchmarks.

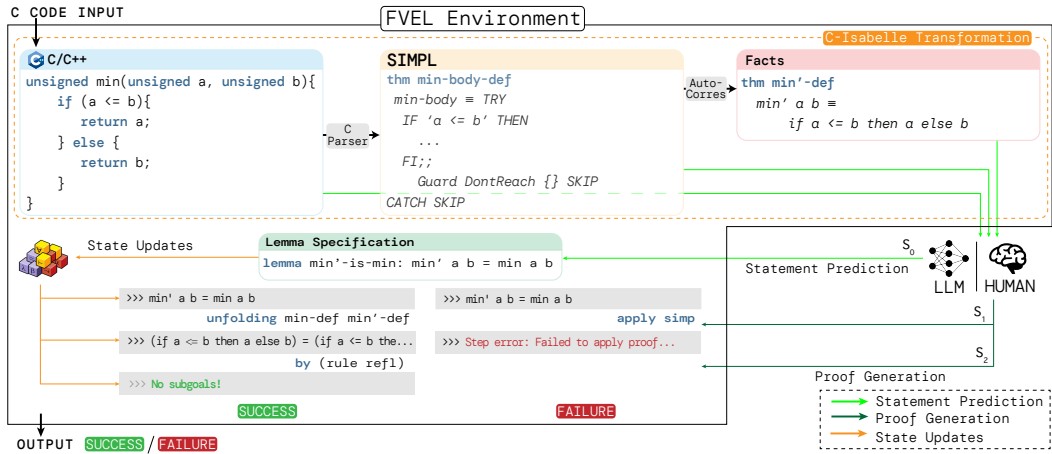

Figure 1: FVEL workflow. FVEL takes a C code as input, parses it into Isabelle definition, and then conducts interactive formal proving with FVEL-LLM/human via outputting proof state and receiving generated proof.

libraries with a large number of human-written and checked theorems and rules, which are provided as pre-training materials for many large language models [27, 37, 14]. The ATP formulation and rules have strong expressiveness and, therefore have a great potential for describing formal verification problems and requests. As a result, the verification can be implemented under a rigorous, step-wise, and interactive ATP environment. Moreover, the pre-trained formal reasoning capabilities within LLMs and their potential to solve formal verification problems are underexplored.

To take one step towards this goal, this paper proposes FVEL, a new formal verification environment interacting with LLMs via automated theorem proving processes. Figure 1 demonstrates an overview of FVEL. Specifically, the FVEL environment takes as input a code to be verified, converts the code into Isabelle formulation, and generates a lemma in Isabelle followed by a whole proof to the lemma. FVEL then outputs the proof result (succeed or failed being proved) as an indication of the code verification result. FVEL interacts with an LLM by initially providing the converted Isabelle formulation to the LLM and then receiving the derived lemma on the code specification. The interaction is then continued by the LLM generating proof states and the FVEL environment providing feedback via prover information in the PISA environment [15], such as cheating keywords `sorry` or `opps` and other error messages. As a result, a user provides her code to be verified to FVEL, and then she will receive the verification result and intermediate proving information. Note that we follow previous works [6, 40] to investigate FVEL on C code verification in this paper. We remain the extension of FVEL to support more programming languages as a near future work.

To implement the FVEL environment, we extract and cleanse a large-scale FVELER dataset with deep dependencies, which can be applied as both a fine-tuning resource and evaluation benchmark. The FVELER dataset has two main components: C code dependencies formulated by Isabelle theories, and Isabelle lemmas with their step-wise proof states. FVELER then includes 758 theories with 29,304 lemmas and 201,498 proof steps. The dataset is then randomly split according to lemmas, resulting in training/validation/test/test-hard sets. The test-hard set data have dependencies that are challenging to find. Statistical analysis shows that FVELER data comprehensively covers diverse dependency depths and has a remarkable number of data with very deep dependencies. For example, over 50% of lemmas have a depth greater than 78, while the deepest dependency is 156.

We then benchmark FVELER in the FVEL environment on the Code2Inv [36] and SV-COMP [2] benchmarks. After fine-tuning on FVELER, Mistral-7B [14] and Llama3-8B[4] are observed performance improvements on both benchmarks. For example, Llama3-8B solves 81 out of 1,000 SV-COMP problems, achieving a 17.39% improvement, and Mistral-7B improves by 12%. Moreover, ablation study on statement and proof errors during FVEL verification shows that after fine-tuning with FVELER, the proportion of proof errors is reduced, indicating the benefits of FVEL and FVELER. The contributions of this paper are summarized as follows:

---

[4] https://github.com/meta-llama/llama3

1. We introduce FVEL, an interactive formal verification environment with LLMs that leverages neural ATP advances including formulation, theorems, models, and prover.

2. We extract and cleanse a large-scale FVELER with 758 theories, 29,304 lemmas, and 201,498 proof steps in total that contain deep dependencies. We split FVELER into training/validation/test/test-hard sets as fine-tuning resources and an evaluation benchmark.

3. We apply FVEL with several FVELER fine-tuned LLMs. The results show that FVEL with FVELER fine-tuned LLMs show performance improvements on representative code verification benchmarks, and the proof errors are reduced. The results indicate the benefits of FVEL and FVELER.

## 2 Related Works

**Formal Verification.** Formal verification (FV), or automated program verification [36, 2], is the task of verifying if a given code fulfills specific requirements. One line of work [10, 6, 21] resort to reducing the code into candidate loop invariant and then using satisfiability modulo theories (SMT) solver for post-hoc verification. Different methods are proposed to improve the loop invariant inference, including decision tree [20], reinforcement learning [42, 35], and neural network [33]. However, finding or generating accurate loop invariants remains challenging, which hinders the preciseness of the verification. Moreover, symbolic SMT solvers are time-consuming and uneconomical when there is a large amount of code to be verified. The other line of work tries to introduce LLMs to solving formal verification. For using LLMs to find loop invariants, Loopy [17] prompts LLMs to exhaustively generate candidate invariants and include a repair procedure to improve the variants by an SMT solver. For using LLMs to perform the program verification, Lemur [40] proposes to integrate LLMs in formal verification by transforming the program invariants into deductively verified sub-goals, appearing to be most relevant to our work. However, they hand-craft a proof system with solely 8 rules without a demonstration of its completeness. Therefore, the expressiveness of this hand-craft system is unclear. In this paper, we propose a new formal verification environment that interacts with large language models to leverage their theorem-proving ability and also the rigorous validation by automated theorem provers. The environment thus leverages the corresponding extensive rule and theorem libraries.

**Automated Theorem Proving with LLMs.** The field of automated theorem proving (ATP) [34, 19, 3, 28, 22] utilizes formal languages such as first-order logic (FOL) and higher-order logic (HOL) to describe mathematical problems, theorems, and solution processes, allowing rigorous step-by-step validation through deductive reasoning to achieve final answers or proofs. Interactive theorem proving (ITP) then introduces interactive proof assistants [29, 5, 4, 25] and automates the validation process with machine learning methods [30, 9, 16, 39]. Furthermore, recent studies explore the integration of large language models and theorem proving [15, 41, 38, 12, 23]. For example, PISA [15] introduces an environment that allows language models to interact with an Isabelle server, which are able to mine 183k lemmas and theorems from the Isabelle libraries. LeanDojo [41], on the other hand, is a Lean environment that enables interaction between the language models and the Lean prover with fine-grained annotations of premises in proofs and an LLM-based theorem prover. Such interactive proving systems leverage both the abundant libraries of theorems and rules and advanced performances of LLMs, which is promising for formalized applications such as formal verification. To this end, this paper investigates a novel LLM interactive environment that advances formal verification. The environment thus also helps solve automated theorem proving tasks.

## 3 FVEL: Interactive FV Environment with LLMs

**Workflow of FVEL.** Figure 1 demonstrates FVEL. The main idea of FVEL is to provide an interactive environment with large language models (LLMs) that leverage rigorous theorem-proving processes. The input of FVEL environment is a code to be verified. Specifically, we follow previous studies [10, 40] to verify C code and conduct a pilot study on our new framework. Moreover, the input format is flexible as one can choose to input an ensemble of C code and its corresponding SIMPL and/or Isabelle content as supplements. The output of FVEL is the code verification result, i.e., success or failure.

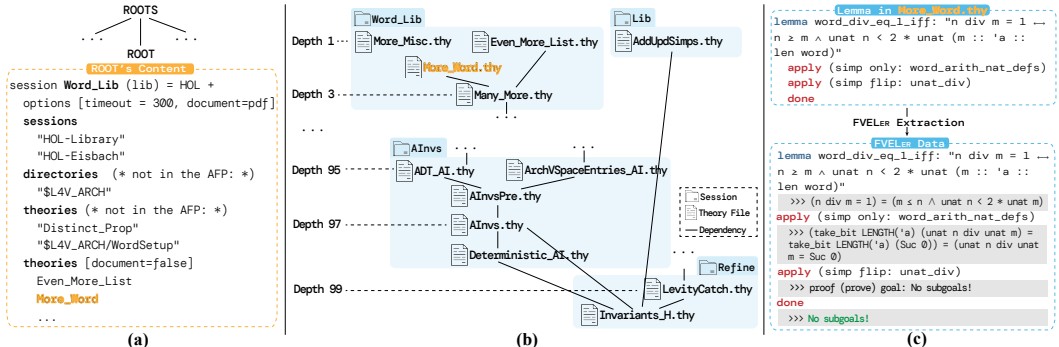

Figure 2: (a) SeL4 ROOT file structure. It provides an example ROOT file content for the session `Word_Lib`. (b) Theory dependency graph. Each theory file is grouped by the Session. (c) Step-wise lemmas extraction.

FVEL interacts with a large language model to achieve the verification. At the initial step of interaction ($S_0$ in Figure 1), FVEL transforms the input C code into facts, and then provides the facts to the LLM. The LLM then generates a lemma in Isabelle [29] as a formal description of the code specification. In this step, a code verification problem is transformed into an ATP problem. As a result, FVEL can leverage the LLMs theorem-proving techniques and rigorous ATP validation. At the follow-up interaction steps ($S_i, i \geq 1$ in Figure 1), the LLM is prompted to generate proof steps, while FVEL incorporates an Isabell prover to provide feedback such as error messages to the LLM. The process terminates until a whole proof is generated. If the proof success in proving the lemma, FVEL outputs "success", otherwise outputs "failure".

**Applying FVEL.** The current version of FVEL supports code verification in C language. We leave the generalization of FVEL to other programming languages as a near future work. To apply FVEL, a user prepares her C code and passes it to FVEL. The user can customize her LLM for FVEL. Therefore, FVEL adjusts to cutting-edge LLMs with strong theorem-proving ability and customized LLMs. The user then gets the "success" or "failure" feedback regarding the verification result from FVEL. Furthermore, the intermediate proof states and prover messages provide further information about the verification.

**Environment Implementation.** We perform the C code transformation with the C-Parser [26] and AutoCorres [7] and construct the environment based on Isabelle-scala[5] and PISA [15]. C-Parser can translate a large subset of C99 code into the imperative language SIMPL. For every function in the C source file, it generates a corresponding Isabelle definition literally without omitting details of the C language. AutoCorres can further simplify and abstract the generated SIMPL language, producing a higher-level functional specification that is easier to reason by humans. We provide the simplified Isabelle definition to LLMs to better align with human interactive proving with Isabelle. Specifically, Given the C source file, we use the PISA to set up the Isabelle process by including the directories of C-parser and AutoCorres in the Isabelle "sessionRoots", and setting the "`workingDirectory`" to the C file. Then we initialize the Isabelle state by importing the C-parser and AutoCorres tools. Lastly, we use PISA to interact with the Isabelle process, invoke tools to translate the C code, and then extract the fact definition "c file name.function name'_def" after unfolding it in Isabelle. The extracted definition can be passed to LLMs, and LLMs can generate lemma specifications and interact with Isabelle prover in this setup process.

## 4 FVELER: Benchmarking FVEL

### 4.1 FVELER Overview

FVELER contains transformed Isabelle theories and lemmas from C codes that support the FVEL environment for C code verification. FVELER has two main components: (1) Theories dependencies. A resource for dependencies among theories, lemmas, and C code specified by SeL4 verification. These data provide the ground-truth seL4 premises for proving the current lemma and enable a

[5] https://github.com/dominique-unruh/scala-isabelle

model to retrieve related statements or proof context at both the training and testing stages. (2) Lemmas from theories with their Isabelle proof states. The step-wise lemmas with multiple proof states that support the Isabelle proving process in FVEL. These data on the one hand enhance LLMs with search-based/step-wise ATP while interacting with FVEL, and on the other hand, provide a benchmark for interactive formal verification. Figure 2 illustrates the construction processes of each component.

In the following, we first introduce the preliminary for FVELER construction (Section 4.2), then introduce the construction of the two components in FVELER: (1) the extraction of C-Code Dependencies by Isabelle Theories (Section 4.3) and (2) the extraction of step-wise lemmas (Section 4.4). We then demonstrate FVELER statistics and distribution in Section 4.5.

## 4.2 Preparation

**Data Source.** SeL4[6] [18] is a system microkernel with comprehensive formal verification. Its implementation verification against safety and security specifications contains multi-level formal proof manually written in Isabelle, including abstract specification and concept level to concrete implementation level. Since the open-source seL4 verification contains high-quality and multi-level proof following human reasoning, we choose seL4 as FVELER data source. Figure 2(a) demonstrates the relations amount session, ROOT files, and lemmas in seL4.

**SeL4 Session.** In seL4, an Isabelle session contains a group of theory files that focus on proofing one concept or topic, similar to the package in a programming language. Since the formal verification of seL4 is a large project that involves various aspects, different sessions are used to define code specifications, construct intermediate definitions, and process C code semantics. Isabelle can build a session into a binary file called "heap image" that can be fast-loaded for processing other theories.

**ROOT Files.** The ROOT files contain all the listed ROOTs that should be built by Isabelle. ROOT files instruct Isabelle on how to build the sessions and verify the theories. Each session in a ROOT file contains its names, parent sessions, entry theories, and directories of theory files. We use such information to recursively construct the dependency graphs and set up the Isabelle environment to extract step-wise proof states.

**Theory and Lemma.** A Theory file contains the necessary context and concrete proof for Isabelle to formally verify the target lemmas. The context includes importing other theories, defining intermediate symbols, and giving concrete lemma statements and proof. A lemma is a statement that relates to the functionality demands of the codes. In FVEL, the goal of formal verification is to generate the correct proof of these lemmas.

## 4.3 FVELER Construction: C-Code Dependencies by Isabelle Theories

The dependencies are all formulated and saved in Isabelle. The extraction of the dependencies is via constructing a theory dependency graph. Figure 2(b) illustrates the theory dependency graph. This graph nodes are the `.thy` theory files in seL4 while the edges are the `import` relationships between the theory files. It traces multi-hop dependency relationships by `import` among the Isabelle lemmas within the theory files. With the theory dependency graph, it is convenient to locate and extract multi-depth lemmas and their corresponding proofs.

While constructing the dependency graph, we first traverse all ROOT files according to the file order specified in seL4, and then parse the session and corresponding theories recursively to obtain the dependency relationship. Specifically, the graph construction is started by sequentially parsing the ROOT files in the seL4 ROOTS file. For each session, we match the keywords to extract its name, its parents, and its directories. After extracting all session information, we traverse the ROOTS again and parse the theory files under the "theories" keywords. We parse the string between "imports" and "begin" keywords to extract the dependency relationships of these parent theories and parse these theories recursively to form a graph of other theories given current or other session information. After traversing all sessions, we construct a dependency graph among sessions and theories, which can be used to provide dependent proof context or premise when generating formal verification.

---

[6]The l4v library which contains the proofs for the SeL4 kernel are licensed under GPL version 2.

Table 1: FVELER Statistics. A *theory* is a `.thy` file in seL4 that contains multiple *lemmas*. Each *lemma* has multiple *proof steps*. The train/val/test/test-hard data split is based on *lemmas*.

| | Total | Train | Val | Test | Test-Hard |
|---|---|---|---|---|---|
| ▷ *Theory* | | | | | |
| Number of Theories | 758 | - | - | - | - |
| Average depth* | - | 73.687 | 73.732 | 73.958 | 31.476 |
| Maximum depth | 156 | 156 | 155 | 155 | 115 |
| ▷ *Lemma* | | | | | |
| Number of Lemmas | 29,304 | 26,192 | 1,145 | 1,115 | 852 |
| ▷ *Proof Step* | | | | | |
| Number of proof steps** | 201,498 | 181,887 | 6,931 | 8,036 | 4,644 |
| Average proof steps | - | 6.944 | 6.053 | 7.207 | 5.450 |
| Maximum proof steps | 963 | 963 | 188 | 574 | 107 |

\* Depth: Degree of the theory dependency graph by `import` relationship.
\*\* Proof step: A single step in Isabelle producing a valid statement for interaction."

## 4.4 FVELER Construction: Step-Wise Lemmas

For extracting the lemmas and also saving their dependencies by theory files and their proof states, we leverage the PISA [15] environment. We initial the PISA environment and parse all theory files based on the session information the theory dependency graph developed in Section 4.3. Specifically, As shown in Figure 2(c), we first build the seL4 formal verification project[7] and obtain the sessions' binary heap images. Then given each theory file, we modify the PISA environment to include and load all dependent sessions, setting the working directories to the processed theory files, and then temporarily copying the files from session directories to the current one, such that the Isabelle process can correctly import all dependent theory. Lastly, we use PISA to parse the theory file into multi-step and perform step-wise interaction with Isabelle. For each step, Isabelle will return a proof state and we store the step and proof state as a step-wise training sample. We traverse the seL4 verification projects and extract most of the theory files. Specifically, we omit some experimental theory files that can not be verified by Isabelle or failed when interacting with PISA. We also omit the sessions for documentation, C parser [26] and AutoCorres [7] as they do not contain lemmas that are relevant to formal proving.

## 4.5 FVELER Splits, Statistics, and Distributions

**Splits.** We randomly split FVELER according to lemmas, resulting in a training set, a validation set, a test set, and an especially selected test-hard set. The test-hard set is selected from those lemmas in the three sessions "SysInit", "SysInitExamples", "LibTest". Such lemmas are in higher depths in the dependency relationship, therefore they have less `import` relationships by other theories.

**Statistics.** Table 1 demonstrates the number of samples in FVELER and each data split. FVELER in total contains 758 Isabelle theories, with 29,304 lemmas and 201,498 proof steps. The average dependency depths among the theories range from 31 to 73. The maximum dependency path reaches a depth of 156. The average proof step ranges from 5 to 8, while the maximum of proof steps in a lemma reaches 963. In general, FVELER is a large-scale dataset with deep dependencies among the Isabelle theorems and lemmas that fit C code formulation. It thus supports the interactive C code verification with a theorem-proving LLM.

**Distribution of Dependency by Theory.** We quantify the dependency by "depth", which is the degree of the theory dependency graph by the `import` dependency relationship among the theory files, as introduced in Section 4.3. Figure 3a demonstrates the distribution of theories by the depth of dependency relationship. Besides the number of theories in depth=1 is the highest 59 followed by depth=2 and depth=3 with 29 theories, respectively, small peaks are observed in multiple depth levels. For example, depth=7 has 19 theories, depth=16 to 22 have around 15 theories, and there are still 11

---
[7] https://github.com/seL4/l4v

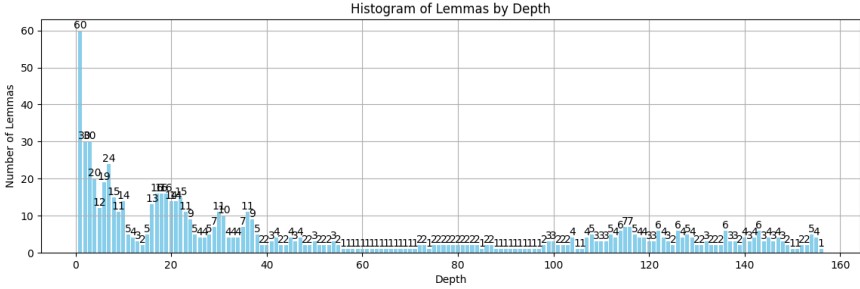

(a) Distribution of dependency by theory.

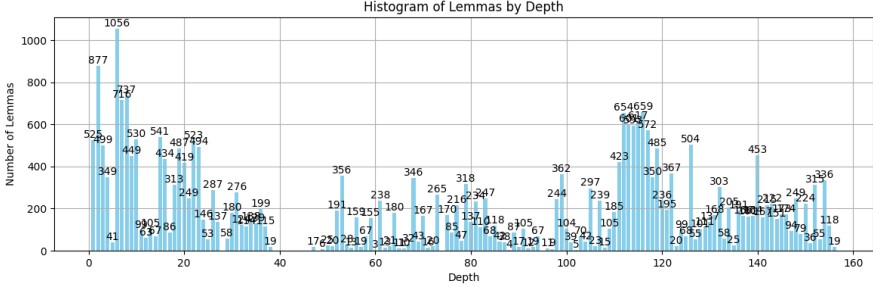

(b) Distribution of dependency by lemma.

Figure 3: The FVELER dependency distributions by theory and lemma, respectively.

theories that have depth=36. Most impressively, depth=112 to 115 appear to have on average around 10 theories. As a result, FVELER has very in-depth and comprehensive dependencies information, which can be beneficial for not only code verification with dependencies but also multi-step ATP.

**Distribution of Dependency by Lemma.** Figure 3b demonstrates the distribution of 29,304 lemmas by depth. That is, each lemma belongs to one of the 758 theories whose depth in its dependency is calculated here. Therefore, in Figure 3b we observe a more fine-grained dependency distribution within the theory files. It is shown that lemmas with deep dependency are widely distributed. Lemmas with depth≥78 are 14,668, over 50% of all lemmas. For example, depth=116 there are 659 lemmas. Moreover, there are also 11,518 lemmas with shorter depth=1 to 40. Besides, a curious observation is that depth=39 to 46 are not found in lemmas. Therefore, FVELER widely supports verification with diverse depths of dependency.

**Distribution of Lemma Steps.** One proof step in a lemma is from a current proof state to the next which produces a sound statement for interaction in PISA. Figure 4 demonstrates the distribution of intermediate proof steps of the 29,304 lemmas. It is indicated that the number of proof steps are dramatically different amount the lemmas. 12,089 out of the 29,304 lemmas can be proved via one proof step. Proof steps between 2 and 10 there are 12,957 lemmas. Therefore, over 85% of the lemmas in FVELER can be proved within 10 steps. Moreover, 28,954 out of the 29,304 lemmas can be proved within 100 steps. Therefore FVELER is more helpful for verification within 100 ATP steps, which is sufficient for covering most of the cases in practice.

## 5 Benchmark Study

### 5.1 Setup

**Dataset.** We benchmark FVELER in the FVEL environment on Code2Inv [36] and SV-COMP [2]. The Code2Inv dataset contains 133 programs in C, and the SV-COMP dataset is from the Software-Verification Competition with over 23k C programs. These two datasets are purposed for formal verification. SV-COMP is a competition to establish a set of benchmarks for software verification. In the competition, the verifiers input a program that implements a particular function and a specification that describes the expected behavior of the program, e.g., no illegal accesses to arrays, no memory

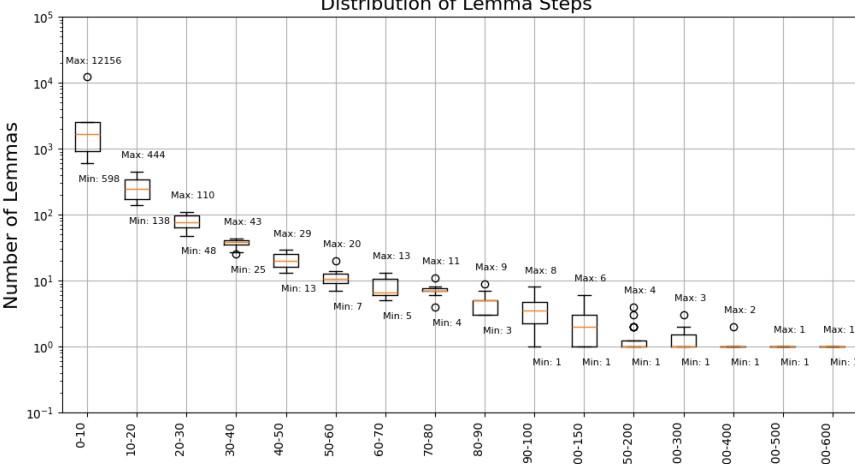

Figure 4: The FVELER lemma distribution over step intervals. The y-axis is adjusted to a logarithmic scale.

leaks, and so on. The verifiers judge that the program meets the specification and output a boolean value. Code2Inv covers a wide range of programs with one or more nested and conditioned loops, and the verifiers are required to find out the loop invariants during the loop, which is the key to understanding the work logic of the function. In this task, we convert the problem into proving that the loop invariant satisfies the loop. Since C-parser supports only part of the C99 standard, we normalize the C code to make C-parser work properly. Please refer to the supplementary materials for more details on preprocessing and implementation.

**Fine-tuning.** We use the training set of FVELER to fine-tune language models. In this study, we employ LORA [11] to fine-tune two most advanced open-source large language models which excel in mathematical reasoning and code generation: Llama-3-8B-instruct[4] and Mistral-7B-Instruct-v0.2 [14]. We convert the training data into the alpaca format, where all training samples use the same instruction, the input is the lemma specification, and the output is a complete proof written in Isabelle.

**Inference.** During inference, we transfer the input c-code functions into Isabelle facts in FVEL environment, requiring the language model to generate a lemma specification to verify that it satisfies the specifications (e.g., that the assertion holds or does not result in an overflow). The language model generates proof and interacts with PISA. If proof is passed by Isabelle proving environment, we consider it a successful verification.

**Evaluation.** We follow the evaluation settings of Lemur [40]. Within a specified timeout, Lemur, UAotumizer, and ESBMC generate proposals and call solvers for verification. Our approach interacts with PISA and self-corrects by the returned error messages.

### 5.2 Compared Methods

The methods we compare include the symbolic solvers: Uautomizer [10] and ESBMC [6], and the LLM-based method: Lemur [40]. UAUTOMIZER [10] is the overall champion of the 12th Competition on Software Verification (SV-COMP 2023)[8]. Combined with static analysis and model checking, it is one of the few verifiers that can give witness during verification. ESBMC [6] is based on K-induction, which is particularly useful for verifying the properties of loops and recursive functions. Lemur presents a set of derivation rules and makes proposals using a language model to approximate the boundary conditions of the loop invariant by interacting with the verifier.

### 5.3 Formal Verification Results

Table 2 reports the number of passed verification tasks. Formal verification for C code in the Isabelle environment is a great challenge. First, the language model needs to generate the correct lemma

---

[8]https://sv-comp.sosy-lab.org/2023/

Table 2: Result on formal verification task. FT: Fine-tuned.

| Model | Code2Inv (#=133) | SV-COMP-47 (#=47) | SV-COMP (#=1,000) |
|---|---|---|---|
| ▷ *Symbolic Solver* | | | |
| UAUTOMIZER [10] | 92 | 1 | 374 |
| ESBMC [6] | 68 | 1 | 358 |
| ▷ *LLM-based Solver* | | | |
| Lemur-GPT-3.5-turbo [40] | 103 | 14 | - |
| Lemur-GPT-4 [40] | 107 | 25 | - |
| Mistral-7B [14] | 37 | 10 | 75 |
| Mistral-7B-FT | 40 | 14 | 84 |
| Llama3-8B[4] | 46 | 11 | 69 |
| Llama3-8B-FT | 46 | 16 | 81 |

Table 3: Failure types of Code2Inv and SV-COMP datasets.

| Model | Code2Inv | | SV-COMP | |
|---|---|---|---|---|
| | statement error (%) | proof error (%) | statement error (%) | proof error (%) |
| Mistral-7B | 70.8 | 29.2 | 49.7 | 50.3 |
| Mistral-7B-FT | 72.0 | 28.0 | 59.0 | 41.0 |
| Llama3-8B | 67.8 | 32.2 | 53.0 | 47.0 |
| Llama3-8B-FT | 66.7 | 33.3 | 61.8 | 38.2 |

specification, which is particularly difficult on code2inv and SV-comp-47 datasets with loops or complex conditions, and thus our fine-tuned prover model achieves limited performance gains. On the Code2Inv dataset, the uncertain looping conditions pose an additional challenge for the language model to validate C programs. The performance of the fine-tuned model on the SV-COMP-47 dataset equals or exceeds that of Lemur-GPT-3.5-trubo. In addition, symbolic solvers overwhelmingly dominate the SV-comp-1,000 dataset, which covers diverse specifications. The lack of a relevant corpus makes it difficult for language models to verify specifications such as concurrency and no-overflow. Since FVELER originates from the seL4 micro-kernel operating system, the correlation of the data makes fine-tuning on the SV-COMP dataset effective.

## 5.4 Ablation Study

Further analysis in Table 3 shows that most of the validation errors in the Code2Inv dataset come from specification generation, which can be type mismatching, syntax errors, etc. In particular, it is difficult to generate an accurate lemma specification under uncertain loop conditions. In contrast, the SV-COMP dataset has a larger fraction of validation errors from proof generation, and our finetuned prover model effectively reduces these proof errors. It suggests that it is feasible to utilize language models for formal verification in the Isabelle environment, but how to verify that the lemma specification generated by the model is semantically and syntactically correct remains a challenge.

## 5.5 Generalization to Other Programming Language

Given Python's status as one of the most widely adopted programming languages, it is frequently utilized in code generation and tool invocation for Large Language Model inference. This study contemplates the formal verification of Python code. However, many existing code translation datasets are deficient in aligned C to Python samples. The C code prevalent in most datasets is written in C++, which poses a challenge since the C-parser tool accommodates only the C99 standard. Considering the complexities associated with employing automated tools or rules to normalize these C++ codes to C99, we have elected not to leverage open-source datasets such as HumanEval-X [44], CodeNet[31], and others. Instead, we have collected algorithmic solutions implemented in various programming

Table 4: Result on Python (Translated to C) Code Verification.

| Model | # Verified |
|---|---|
| Mistral-7B | 35 / 93 |
| Mistral-7B-FT | 42 / 93 |
| Llama3-8B | 38 / 93 |
| Llama3-8B-FT | 43 / 93 |

languages from the Online Judge platforms (LeetCode[9] and POJ[10]) and manually ascertained their semantic equivalence. Consequently, we have compiled a modest test dataset comprising 93 samples, with each data point featuring both a Python solution and a corresponding C solution. An exemplar data point is presented in the supplementary materials. This dataset will be made accessible on our GitHub repository.

We employed GPT-4 to translate the collated Python code into C. Through prompt engineering, we ensured that GPT-4 generated code adhering to the C99 standard and evaluated the quality of the generated code using the success rate of C-parser parsing and the CodeBLEU[32] metric relative to the ground truth. The results indicate that GPT-4 achieved an 84% pass rate (79/93) and a CodeBLEU score of 81.32.

Subsequently, we input the translated C code into our framework and conducted verification using our baseline models. The outcomes are delineated in Table 4.

After fine-tuning, Mistral and Llama successfully validated 42 and 43 pieces of code, respectively. This indicates that pre-trained Large Language Models have already demonstrated significant capability in verifying simple Python programs that do not involve complex import structures. With additional fine-tuning, performance can be further enhanced. Moreover, the FVELframework has the potential to be adapted for the verification of code in other programming languages, showcasing its versatility and applicability beyond the current scope.

## 6    Conclusion

This paper proposes FVEL, an interactive formal verification environment that can interact with LLMs by formulating formal verification (FV) dependencies and requests into automated theorem proving (ATP) theories and lemmas, and the verification processes into lemma proofs. We extract and cleanse a large-scale dataset FVELER with deep dependencies among Isabelle theorems and lemmas that formulate the formal verification. Statistical analysis suggests that FVELER has comprehensive and deep dependency information among the theorems and lemmas, and the multi-step lemma proofs reach 100 steps. We benchmark FVELER by fine-tuning LLMs and then interacting with the FVEL environment. We evaluate Llama3-8B and Mistral-7B in this setting. Evaluations on Code2Inv and SV-COMP show improvements. For example, performances on SV-COMP of 17.39% (69→81) by Llama3-8B and 12% (75→84) by Mistral-7B, and the proof error proportions are reduced. Experiments on Python code show that our approach has the ability to generalize to validate other programming languages. The results demonstrate the benefits of FVEL and FVELER.

## Acknowledgements

This work was supported in part by the National Science and Technology Major Project (2020AAA0109704), National Science Foundation of China Grant No. 62476293, Guangdong Outstanding Youth Fund (Grant No. 2021B1515020061), Shenzhen Science and Technology Program (Grant No. GJHZ20220913142600001), Nansha Key RD Program under Grant No.2022ZD014, the Major Key Project of PCL (No. PCL2024A04), China Postdoctoral Science Foundation No. 2023M744001.

---

[9]https://leetcode.com/
[10]http://poj.org/

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
