# FVEL: Interactive Formal Verification Environment with Large Language Models via Theorem Proving

Xiaohan Lin[1]*    Qingxing Cao[1]*    Yinya Huang[2]*    Haiming Wang[3]*    Jianqiao Lu[4]

Zhengying Liu[5]    Linqi Song[2]    Xiaodan Liang[1,6,7]†

[1]Shenzhen Campus of Sun Yat-sen University    [2]City Univeristy of Hong Kong
[3]Sun Yat-sen University    [4]The University of Hong Kong
[5]Huawei Noah's Ark Lab    [6]MBZUAI    [7]DarkMatter AI Research

# Appendix

---

\*   Equal contribution.
†   Corresponding author.

Submitted to the 38th Conference on Neural Information Processing Systems (NeurIPS 2024) Track on Datasets and Benchmarks. Do not distribute.

# A  Limitations

In this work, we follow previous works [4, 2, 8] to test FVEL on C code verification. We remain the extension of FVEL and the corresponding FVELER to support more program languages as a near future work. Additionally, semantic alignment between lemma statements and program specifications is an unexplored area of research.

# B  Societal Impacts

The research presented in this paper has the potential to advance the field of formal verification, automated theorem proving, AI for Math, and software engineering. The advancement can enhance the capabilities of large language models in formal verification, contributing to more reliable software development. By directly releasing the code and data, we aim to ensure the responsible use of our work, fostering further innovation and maintaining high standards of data privacy and intellectual property compliance. The proposed FVEL and FVELER benchmark the interactive formal verification performance in the machine learning field. Therefore we claim that there are no negative social impacts in this paper.

# C  FVELER Benchmark

## C.1  Dataset Format

We first list the folder and files under the FVELER directory. We then demonstrate the detailed formats of the folder/files.

- `sel4_extraction/` is a folder that has the same structure as the sel4 verification project (l4v). Each file is the extracted step-wise proof state of the corresponding l4v theory files. For example, "sel4_extraction/proof/invariant-abstract/AInvs.json" is the proof state of the file l4v/proof/invariant-abstract/AInvs.thy.

- `dataset_lemma_split.json` contains all lemmas proof steps and states, and splits them into the train, val, test, and test-hard set.

- `sel4_thy_info.json` contains information of all theory files, including their names, dependency relations, and lemmas.

- `sel4_session_info.json` contains all session information, including dependent sessions, theories, and directories.

### C.1.1  `sel4_extraction/`

The `sel4_extraction/` folder contains parsed l4v theory files. Each theory file in this folder is a JSON file, storing a list of whole proof steps, and each step is stored as a dictionary. The file structure and a sample proof step are demonstrated as follows:

```
sel4_extraction/proof/invariant-abstract/AInvs.json:
[
  ...,
  {
    "index": 2,
    "step": "lemma st_tcb_at_nostate_upd: ...",
    "raw_output": "proof (prove)\ngoal (1 subgoal)...",
    "step_time": 0.11420297622680664
  },
  ...
]
```

Each proof step dictionary has the following fields:

- "`index`": The index of this step.

68  • "step": The proof step in Isabelle.

69  • "raw_output": The returned proof state in Isabelle.

70  • "step_time": The processing time of this step.

71 ## C.1.2  `dataset_lemma_split.json`

72 The `dataset_lemma_split.json` file stores the train/val/test/test-hard splits. Each split is a list of
73 lemmas, and each is stored as a dictionary. The file structure and a sample lemma are demonstrated
74 as follows:

```
76 {
77   "train": [
78     {
79       "context": "lemma n_less_equal_power_2:\n  \"n < 2 ^ n\" by (
80           fact less_exp)",
81       "proof": [
82         "lemma n_less_equal_power_2:\n  \"n < 2 ^ n\"",
83         "by (fact less_exp)"
84       ],
85       "proof_state": [
86         "proof (prove)\ngoal (1 subgoal):\n 1. n < 2 ^ n",
87         ""
88       ],
89       "statement": "lemma n_less_equal_power_2:\n  \"n < 2 ^ n\"",
90       "theory_name": "More_Arithmetic",
91       "num_steps": 1
92     },
93     ...
94   ],
95   "val": [ ... ],
96   "test": [ ... ],
97   "test-hard": [ ... ]
98
99 }
```

100 Each lemma dictionary has the following fields:

101  • "context": Full lemma context in plain text.

102  • "proof": A list of all proof steps in Isabelle.

103  • "proof_state": A list of all proof states in Isabelle.

104  • "statement": The lemma statement to be proved.

105  • "theory_name": The name of the theory where this lemma belongs.

106  • "num_steps": The number of steps for proving this lemma.

107 ## C.1.3  `sel4_thy_info.json`

108 `sel4_thy_info.json` contains information regarding the theory files, stored as a dictionary where
109 a key is a theory file and the value contains the related information. A sample is demonstrated as
110 follows:

```
112 {
113   ...,
114   "/lib/Word_Lib/More_Word.thy": {
115     "name": "More_Word",
116     "dependency": {
117       "HOL-Library.Word": "",
118       "More_Arithmetic": "/lib/Word_Lib",
119       "More_Divides": "/lib/Word_Lib",
120       "More_Bit_Ring": "/lib/Word_Lib"
121     },
122     "depth": 2,
123     "related_c_code": [],
```

```
124      "child": [
125        "/lib/Word_Lib/Aligned.thy",
126        "/lib/Word_Lib/Bit_Shifts_Infix_Syntax.thy",
127        ...,
128        "/lib/Word_Lib/Machine_Word_64.thy"
129      ],
130      "path": "/lib/Word_Lib/More_Word.thy",
131      "session": "Word_Lib",
132      "lemmas": [
133        {
134          "context": "lemma sofl_test: ...",
135          "proof": [...],
136          "proof_state": [...],
137          "statement": "...",
138          "theory_name": "More_Word",
139          "num_steps": 25
140        },
141    },
142    ...
143  }
144
```

The information dictionary of a theory file (e.g., "/lib/Word_Lib/More_Word.thy") has the following fields:

- "name": The theory name.
- "dependency": A dictionary of dependent theories and their paths. The key is the theory name and the value is the path. A theory that belongs to another session has no path. For example, "HOL-Library.Word" is imported from session "HOL-Library", and its path is empty.
- "depth": The depth of this theory.
- "related_c_code": The C code files called by this theory or any of its ancestors.
- "child": The theory files depending on this theory.
- "path": The theory file path relative to the l4v folder.
- "session": The session that contains this theory.
- "lemmas": The list of all lemmas in this theory files. Each lemma is stored in a dictionary, which is the same as in "dataset_lemma_split.json".

### C.1.4  sel4_session_info.json

sel4_session_info.json contains information regarding each l4v session, stored as a dictionary where a key is an l4v session and the value contains the related information. A sample is demonstrated as follows:

```
164  {
165    "ASpec": {
166      "dependency": [
167        "Word_Lib",
168        "\"HOL-Library\"",
169        "Lib",
170        "ExecSpec"
171      ],
172      "name": "ASpec",
173      "theories": [
174        "/spec/abstract/Structures_A.thy",
175        ...,
176        "/spec/abstract/Exceptions_A.thy"
177      ],
178      "ROOT_dir": "/spec",
179      "ROOT_relative_dir": "abstract",
180      "additional_dir": [
181        ".",
```

```
182        "ARM"
183      ],
184      "depth": 6
185    },
186    ...
187 }
188
```

The information dictionary of a session (e.g., "`ASpec`") has the following fields:

- "`dependency`": A list of all its dependent sessions' names.

- "`name`": The session name.

- "`theories`": The list of all theory files included in this session, represented by their keys in "`sel4_thy_info.json`".

- "`ROOT_dir`": The directory of this session's ROOT file relative to the l4v folder.

- "`ROOT_relative_dir`": The main working directory of this session relative to "`ROOT_dir`".

- "`additional_dir`": The list of additional directories containing this session's theory files relative to "`ROOT_relative_dir`".

- "`depth`": The depth of this session.

## C.2 Datasheet

We present a datasheet [3] for documentation and responsible usage of FVELER benchmark.

**Motivation.**

- *For what purpose was the dataset created?* The FVELER dataset is created to support the interactive formal verification with large language models. It provides lemmas for formally proofing the correctness of a microkernel system with step-wise Isabelle language and state.

- *Who created the dataset (e.g., which team, research group) and on behalf of which entity (e.g., company, institution, organization)?* It was created by the authors of this paper by extracting and cleansing the data from the sel4 verification project (l4v).

- *Who funded the creation of the dataset?* See the acknowledgments once it is available.

**Composition.**

- *What do the instances that comprise the dataset represent (e.g., documents, photos, people, countries)?* The FVELER dataset consists of dependent theory sessions, theory files grouped by sessions, lemmas from theories, and proof states of the lemmas, all written in Isabelle.

- *How many instances are there in total (of each type, if appropriate)?* The FVELER dataset has 758 theories, 29,125 lemmas, and 200,646 proof steps.

- *Does the dataset contain all possible instances or is it a sample (not necessarily random) of instances from a larger set?* The dataset contains all possible theory files, lemma, and their proof that PISA can extract from the sel4 verification project (l4v) in ARM architecture(excluding C Parser and autocorres tools) released on March 11, 2024.

- *What data does each instance consist of?* Each instance consists of the lemma statement, the proof step, and the corresponding state in Isabelle code.

- *Is there a label or target associated with each instance?* Yes, each instance has a target, the next proof step.

- *Is any information missing from individual instances?* No.

- *Are relationships between individual instances made explicit (e.g., users' movie ratings, social network links)?* Yes, each instance is associated with a theory file, which contains dependent theory files as its premises.

- *Are there recommended data splits (e.g., training, development/validation, testing)?* Yes. We recommend four data splits: a training set with 26,081 lemmas, a validation set with 1,115 lemmas, a test set with 1,077 lemmas, and a test-hard set with 852 lemmas.

- *Are there any errors, sources of noise, or redundancies in the dataset?* The extracted lemma is formally verified by Isabelle and thus has no error or noise. There might exist some redundant proof that is very similar to the others.

- *Is the dataset self-contained, or does it link to or otherwise rely on external resources (e.g., websites, tweets, other datasets)?* The dataset is self-contained.

- *Does the dataset contain data that might be considered confidential (e.g., data that is protected by legal privilege or by doctor-patient confidentiality, data that includes the content of individuals' non-public communications)?* No.

- *Does the dataset contain data that, if viewed directly, might be offensive, insulting, threatening, or might otherwise cause anxiety?* No.

**Collection Process.**

- *How was the data associated with each instance acquired?* The original data contains Isabelle theory files structured with ROOT file. We apply FVELto extract their proof steps and states. The details are described in Section 4 of our paper.

- *What mechanisms or procedures were used to collect the data (e.g., hardware apparatuses or sensors, manual human curation, software programs, software APIs)?* The original data is publicly released in https://github.com/seL4/l4v.

- *Who was involved in the data collection process (e.g., students, crowdworkers, contractors) and how were they compensated (e.g., how much were crowdworkers paid)?* No manual effort was involved in the data collection process.

- *Over what timeframe was the data collected?* The dataset was collected on March 11, 2024.

**Preprocessing/cleaning/labeling.**

- *Was any preprocessing/cleaning/labeling of the data done (e.g., discretization or bucketing, tokenization, part-of-speech tagging, SIFT feature extraction, removal of instances, processing of missing values)?* The original l4v theory file is parsed into step-wise language by Isabelle. We then interact with Isabelle using these steps to obtain the step-wise states.

- *Was the "raw" data saved in addition to the preprocessed/cleaned/labeled data (e.g., to support unanticipated future uses)?* Yes. We store the original seL4 formal verification files used for extraction and record the links between each lemma and its original files.

- *Is the software that was used to preprocess/clean/label the data available?* Yes. We release the codes and environments for extracting seL4 formal proofs.

**Uses.**

- *Has the dataset been used for any tasks already?* We have used the dataset for fine-tuning Mistral-7B and llama3-8B for the FVEL environment. We also use the dataset to evaluate the fine-tuned models.

- *Is there a repository that links to any or all papers or systems that use the dataset?* `https://fveler.github.io/`.

- *What (other) tasks could the dataset be used for?* The dataset can be used for pertaining LLMs for various downstream tasks, such as ATP, MWP, and code generation.

- *Is there anything about the composition of the dataset or the way it was collected and preprocessed/cleaned/labeled that might impact future uses?* The dataset is based on l4v and is extracted with Isabelle 2023. The lemma proof and proof states might be different from future versions of l4v or incompatible with future versions of Isabelle.

- *Are there tasks for which the dataset should not be used?* No.

**Distribution.**

- *Will the dataset be distributed to third parties outside of the entity (e.g., company, institution, organization) on behalf of which the dataset was created?* Yes, the dataset is publicly available on the Internet.

- *How will the dataset will be distributed (e.g., tarball on website, API, GitHub)?* The dataset can be downloaded as a tarball.

- *When will the dataset be distributed?* The dataset has been released and can be downloaded from `https://huggingface.co/FVELer`.

- *Will the dataset be distributed under a copyright or other intellectual property (IP) license, and/or under applicable terms of use (ToU)?* The dataset is distributed under CC BY 2.0. The dataset was extracted from the https://github.com/seL4/l4v and is licensed under GPL version 2.

- *Have any third parties imposed IP-based or other restrictions on the data associated with the instances?* No.

- *Do any export controls or other regulatory restrictions apply to the dataset or to individual instances?* No.

**Maintenance.**

- *Who will be supporting/hosting/maintaining the dataset?* The authors of this paper.

- *How can the owner/curator/manager of the dataset be contacted (e.g., email address)?* Please contact Qingxing Cao at caoqx8@sysu.edu.cn.

- *Is there an erratum?* No.

- *Will the dataset be updated (e.g., to correct labeling errors, add new instances, delete instances)?* Please check `https://https://fveler.github.io/` for any update.

- *If others want to extend/augment/build on/contribute to the dataset, is there a mechanism for them to do so?* Yes. they can use our released data extraction code for extending instances from updated seL4 or other related data sources.

### C.3 Data Hosting, Licensing, and Maintenance

FVELER benchmark is distributed under the CC BY 2.0 license. The data and the corresponding documentation are hosted on Hugging Face at `https://huggingface.co/FVELer`. The codes for data extraction and experiments with the corresponding documentation are released at `https://github.com/FVELER/FVEL`. The model checkpoints are hosted on Hugging Face Hub. Our website of FVEL and FVELER is `https://fveler.github.io/`, which is the entry point for everything related, including future updates or maintenance.

## D Experiments on FVELER Test Set

### D.1 Implementation Details

We use Mistral-7B-Instruct-v0.2[3] [6] and LLama-3-8B-Instruct models[4] to conduct the experiments on FVELER test sets.

**Fine-tuning.** We use the LLaMa-Factory [9] framework to fine-tune two models on a single RTX 3090 GPU. Specifically, we deploy LORA[5] on the q_proj and v_proj modules of both models. We filter out training samples of a length greater than 1024 and feed the remaining samples into the model with a global batch size of 8. The training samples are transformed into the alpaca format, demonstrated in Table 1.

---

[3]`https://huggingface.co/mistralai/Mistral-7B-Instruct-v0.2`
[4]`https://github.com/meta-llama/llama3`

Table 1: Demonstrations of input-target pairs in alpaca format for fine-tuning FVEL-LLMs.

---

**Instruction**

Prove the following lemma statement in Isabelle.
Ensure that the proof is complete, logically sound and free of redundant content. Use appropriate tactics
and lemmas as necessary. Don't explain.

**Input**

lemma pred_restrs_cut: "(∃y. restrs x ⊆ y) ⇒ (∃y. pred_restrs nn restrs x ⊆ y)"

**output**

apply (clarsimp simp: pred_restrs_def split: next_node.split) apply blast done

---

**Instruction**

Prove the following lemma statement in Isabelle.
Ensure that the proof is complete, logically sound and free of redundant content. Use appropriate tactics
and lemmas as necessary. Don't explain.

**Input**

lemma if_x_None_eq_Some: "((if P then x else None) = Some y) = (P ∧ x = Some y)"

**Output**

by simp

---

**Instruction**

Prove the following lemma statement in Isabelle.
Ensure that the proof is complete, logically sound and free of redundant content. Use appropriate tactics
and lemmas as necessary. Don't explain.

**Input**

lemma visit_merge:
assumes tr: "tr ∈ exec_trace Gamma fn" "Gamma fn = Some f"
and wf: "wf_graph_function f ilen olen"
and ns: "nn ≠ NextNode (entry_point f)"
"∀n ∈ set ns. graph n = Some (Basic nn [])"
"converse (reachable_step graph) '' {nn} ⊆ NextNode ' set ns"
and geq: "function_graph f = graph"
and cut: "∀x. NextNode x ∈ set cuts → (∃y. restrs x ⊆ {y})"
"∀n ∈ set ns. (nn, NextNode n) *notin* rtrancl
(reachable_step graph ⋂ {(x, y). x *notin* set cuts})"
shows "visit tr nn restrs = fold merge_opt (map (λn. visit tr (NextNode n)
(pred_restrs' n restrs)) ns) None"

**Output**

proof - note ns = ns[folded geq] note cut = cut[folded geq] have step_after:
"⋀n i. n ∈ set ns ⇒ trace_bottom_addr tr i = Some (NextNode n)
⇒ ∃st. tr i = Some [(NextNode n, st, fn)]
∧ tr (Suc i) = Some [(nn, st, fn)]
∧ trace_addr tr (Suc i) = Some nn
∧ restrs_condition tr restrs (Suc i)
= restrs_condition tr (pred_restrs' n restrs) i" apply (drule exec_trace_non_Call[OF tr], (simp add:
ns)+) apply (frule ns[rule_format], cut_tac tr(2)) apply (frule trace_addr_SomeD, clarsimp) apply (frule
exec_trace_invariant[OF tr(1)]) apply (cut_tac i=i in exec_trace_step_cases[OF tr(1)]) apply (clarsimp
simp: all_exec_graph_step_cases exec_graph_invariant_Cons
upd_vars_def save_vals_def) apply (simp add: pred_restrs[OF tr(1)] trace_addr_SomeI trace_bottom_addr_def
K_def) done have step_after_single:
"⋀n i. n ∈ set ns ⇒ trace_bottom_addr tr i = Some (NextNode n)
⇒ restrs_condition tr restrs (Suc i)
⇒ (∀n' j. n' ∈ set ns → trace_addr tr j = Some (NextNode n')
→ restrs_condition tr (pred_restrs' n' restrs) j → j = i)" apply clarsimp apply (frule step_after,
erule trace_addr_trace_bottom_addr_eq) apply (frule(1) step_after) apply clarsimp apply (drule(2)
restrs_single_visit[OF tr wf _ _ _ _ cut(1)], simp_all) apply (rule not_trancl_converse_step, rule ns)
apply (simp add: cut) done have visit_after:
"⋀n v. n ∈ set ns ⇒ visit tr (NextNode n) (pred_restrs' n restrs) = Some v
⇒ visit tr nn restrs ≠ None" apply (clarsimp simp: visit_eqs) apply (drule_tac i=i in step_after,
simp add: trace_addr_trace_bottom_addr_eq) apply (rule_tac x="Suc i" in exI) apply clarsimp done show
?thesis apply (rule sym, cases "visit tr nn restrs", simp_all) apply (rule fold_merge_opt_Nones_eq)
apply (rule ccontr, clarsimp simp: visit_after) apply (clarsimp simp: visit_eqs) apply (frule
visit_immediate_pred[OF tr wf _ ns(1, 3)]) apply clarsimp apply (frule(1) step_after, clarsimp) apply
(frule(2) step_after_single) apply (drule in_set_conv_decomp_last[THEN iffD1]) apply clarsimp apply (rule
trans, rule fold_merge_opt_Nones_eq) apply (rule ccontr, clarsimp simp: visit_eqs pc_def ball_Un) apply
(simp add: trace_addr_SomeI) apply (subst visit_known, assumption, simp_all) apply clarsimp done qed

---

We use a cosine learning rate scheduler with the maximum $lr = 1e - 4$, minimal $lr = 1e - 5$,
$warmup\_ratio = 0.1$. Fine-tuning lasts up to 10 epochs with an early-stop mechanism at minimal
evaluation loss. For Mistral-7B and Llama-3-8B, the fine-tuning lasts about 30 GPU hours.

Table 2: Inference template for prompting FVEL-LLMs.

```
Prove the following lemma statement in Isabelle.  Ensure that the proof
is complete, logically sound and free of redundant content.  Use
appropriate tactics and lemmas as necessary.  Don't explain.
{statement}
```

Table 3: Automated theorem proving result on FVELER.

|  | FVELER test (#=1077) | FVELER test-hard (#=852) |
|---|---|---|
| Mistral-7B | 26 | 19 |
| FVEL-Mistral-7B | 74 | 49 |
| Llama-3-8B | 39 | 27 |
| FVEL-Llama-3-8B | 88 | 64 |

Table 4: Compared generated proofs by Mistral-7B and FVEL-Mistral-7B.

| Input | Mistral-7B | FVEL-Mistral-7B |
|---|---|---|
| lemma not_minus: "NOT (x - y) = y - x - 1" | assume "NOT (x - y) = y - x - 1" hence "x - y = y - x - 1" by simp hence "x = y - 1" by simp thus "x = y" by contradiction qed | by (simp add: not_eq_complement) |
| lemma and_and_not [simp]: "(a AND b) AND NOT b = 0" for a b :: "'a::len word" | lemma and_and_not[simp]: assumes "a: 'a::len word" and "b: 'a::len word" shows "(a AND b) AND NOT b = 0" proof - from assms have "a AND b = (a AND b) AND (a OR b)" by simp then have "a AND (b AND NOT b) = (a AND b) AND (a OR b)" by simp finally show ?thesis by simp qed | proof - have "(a AND b) AND NOT b = a AND (b AND NOT b)" by (simp add: word_bw_assoc) also have "... = a AND 0" by (simp add: word_and_not) also have "... = 0" by simp finally show ? thesis by simp qed |

**Inference.** During inference, the model generates the complete proof in a single pass using a zero-shot approach, and PISA verifies the resulting proof. We manually add the imports and include theorems needed for the proof to the environment so that they are correctly referenced. We set the temperature to 0.0 during inference to assess the model's greedy performance. The inference template to prompt the model is demonstrated in Table 2.

## D.2 Results

Table 3 illustrates the result of fine-tuning Mistral and Llama3 on our FVELER training set and testing on the FVELER test set and test-hard set. The fine-tuned Llama-3-8B and mistral-7B effectively improve the correctness of the proofs, with FVEL-Mistral-7B and FVEL-Llama-3-8B each achieving a 4.5% improvement (2.4% -> 6.9% and 3.6% -> 8.1%, respectively) on the FVELER test split. On the more complex FVELER test-hard split, 3.5% (2.3% -> 5.8%) and 4.3% (3.2% -> 7.5%) improvement are achieved respectively. Currently, the pass rate for both Mistral and Llama remains relatively low, indicating that the proposed benchmark poses significant challenges for LLMs. The poor results are primarily caused by these two factors: 1) **Data scarcity.** The amount of data available on formal verification is relatively small compared to the data required to train a general LLM. This is a long-standing challenge in the domain of formal mathematics and formal verification. FVELER remedies the issue by incorporating data from formal verification, but we still require much more data for the LLM to perform better on the subject. 2) **Tactic application style.** The majority of proofs are written in a tactic application style. Compared to the declarative style, these codes cannot be understood even by humans without interacting with Isabelle and checking the proof state information given by the formal system. The current whole proof paradigm requires generating the proof in one go without the help of the proof state information, which poses a significant challenge.

Table 5: Comparison of Original and Processed C Code

| Original Code | Processed Code |
|---|---|
| ```c
extern void abort(void);
extern void __assert_fail(const char *,
    const char *, unsigned int, const
    char *) __attribute__ ((__nothrow__,
    __leaf__)) __attribute__ ((
    __noreturn__));
void reach_error() { __assert_fail("0", "
    nested3-2.c", 3, "reach_error"); }

void __VERIFIER_assert(int cond) {
  if (!(cond)) {
    ERROR: {reach_error();abort();}
  }
  return;
}

int main()
{
  unsigned int x = 0;
  unsigned int y = 0;
  unsigned int z = 0;
  unsigned int w = 0;

  while (x < 0x0fffffff) {
    y = 0;

    while (y < 0x0fffffff) {
    z =0;
  while (z <0x0fffffff) {
    z++;
  }
    __VERIFIER_assert(!(z % 4));
  y++;
    }
    __VERIFIER_assert(!(y % 2));

    x++;
  }
  __VERIFIER_assert(!(x % 2));
  return 0;

}
``` | ```c
extern void abort(void);

void VERIFIER_assert(int cond) {
  if (!(cond)) {
    {abort();}
  }
  return;
}

int main()
{
  unsigned int x = 0;
  unsigned int y = 0;
  unsigned int z = 0;
  unsigned int w = 0;

  while (x < 0x0fffffff) {
    y = 0;

    while (y < 0x0fffffff) {
    z =0;
  while (z <0x0fffffff) {
    z++;
  }
    VERIFIER_assert(!(z % 4));
  y++;
    }
    VERIFIER_assert(!(y % 2));

    x++;
  }
  VERIFIER_assert(!(x % 2));
  return 0;

}
``` |

## D.3 Case Study

Table 4 demonstrates compared generated proofs by Mistral-7B and FVEL-Mistral-7B after being fine-tuned with FVELER. The upper row shows a case in which FVEL-Mistral-7B correctly applies the lemma learned from fine-tuning, thus correcting and simplifying the proof. Contrastively, Mistral-7B generates common `not_eq_complement` without considering a reasonable proof strategy, resulting in a failed proof. In the second case, Mistral-7B rewrites the lemma statement into "`assumes`" and "`shows`" statements, according to which gives an incorrect proof. FVEL-Mistral-7B, on the other hand, expands the brackets in the equation and then is able to derive contradiction according to "(b AND NOT b)", and completes the proof via the contradiction of the right-hand side of the equation.

## E Implementations Details on Code2Inv and SV-COMP

This section provides supplementary details regarding the benchmark study in Section 5.

### E.1 Evaluation Datasets

**Code2Inv [7].** The code2inv dataset contains 133 programs in c, each containing a pre-condition, a loop body (while or for statement), and a post-condition. The verifier needs to verify that the post-condition (an assertion) holds. It is worth pointing out that the condition of a loop or branch in the program may be indeterminate.

Table 6: A Python to C data sample.

| Python Code | C Code |
|---|---|
| ```python
def removeDuplicates(nums: List[int]) ->
    int:
    j = 1
    for i in range(1, len(nums)):
        if nums[i] != nums[i - 1]:
            nums[j] = nums[i]
            j += 1
    return j
``` | ```c
int removeDuplicates(int* nums) {
    int numsSize = sizeof(nums) / sizeof(
        nums[0]);
    int j = 1;
    for (int i = 1; i < numsSize; i++) {
        if (nums[i] != nums[i - 1]) {
            nums[j] = nums[i];
            j++;
        }
    }
    return j;
}
``` |

**SV-COMP [1].** The Software-Verification Competition provides a diverse set of benchmarks for formal verification. sv-comp benchmark contains over 23k c programs, which tend to be more complex than those in code2inv, and each program is accompanied by a .yml file to declare its specifications. These specifications cover requirements such as ReachSafety, MemSafety, ConcurrencySafety, NoOverflows, Termination, etc. The verifier is required to determine whether a program satisfies the given specifications. We sampled the SV-COMP benchmark into two subsets: a 47-sample subset sampled by Lemur [8], which contains samples with multiple nested loops, and a 1,000-sample subset which is randomly sampled from the full set. In particular, we exclude samples that contain floating-point type because the C-parser cannot parse them correctly.

### E.2 Pre-processing

Table 5 demonstrates a randomly selected sample before pre-processing (original code) and after pre-processing (processed code). The pre-processing stages are explained as follows.

**Data Preprocess.** Since C-parser supports only part of the C99 standard, some C features (e.g. "goto" statements, side effects in expressions, etc.) are not supported, we normalize the C code to make C-parser work properly. Specially, for C code which includes:

**String Literal and Illegal Function Name.** Functions with string literals are often used to give warnings to the verifier, we remove these functions and keep only "extern void abort(void);" In addition, we fix illegal function names, for example, by removing the underlines at the beginning of the name.

**Assertion and Assumption.** We replace all the "assert(statement);" and "assume(statement);" with "if (not (statement) {return -1;}". Note that all assertions appear in the "main()" function, so the semantics before and after the replacement are equivalent.

**Unknown Condition.** "unknown()" is often used in the Code2Inv dataset as a condition in "while" or "if" expressions, and we add external declarations to this function: "extern int unknown(void);".

### E.3 Fine-tuning and Inference

See Appendix D.1 for fine-tuning and inference details.

### E.4 A Case of Python to C Dataset.

See Table 6.