# OpenReview forum: "FVEL: Interactive Formal Verification Environment with Large Language Models via Theorem Proving"
_NeurIPS.cc/2024/Datasets_and_Benchmarks_Track — NeurIPS 2024 Track Datasets and Benchmarks Poster_

### Official Review · Reviewer_BbdW · 2024-07-25
**Review of Submission 612**

**Rating:** 7
**Confidence:** 3
**Correctness:** The claims appear to be correct.

**Review:**

The authors address an important problem in the accessibility of formal methods.
The methodology appears reasonable and well done and the dataset is a useful contribution.
However, as of this writing, I was not able to locate code or demos for the FVEL environment (the following GitHub is baren: https://github.com/FVELER/FVEL).
Because the FVEL environment is a central contribution of this work, I am willing to adjust my scoring if the authors could help me locate the code or demos.

**Strengths:**

This paper addresses an important need in the formal methods community: the lack of tooling that effectively integrates ML-backed specification generation and theorem-proving.
The large FVELer dataset is another contribution since data scarcity is an issue.
If the code is made available, then this work has good value to ML+FV researchers.

**Additional Feedback:**

N/A

**Clarity:**

The clarity is okay, but some sentences have awkward phrasing and flow. Although there are many, these are generally minor.

**Documentation:**

The FVELer dataset is available, but I could not find the code for FVEL.

**Ethics:**

There are no ethical concerns.

**Limitations:**

The authors sufficiently address the limitations.

**Opportunities For Improvement:**

At present, my biggest concern is that I cannot find the code/demos for FVEL, and I would appreciate it if the authors could help me locate them.
Because the FVEL framework is a central contribution, this work does not feel complete without the code.
Unfortunately, I could not find anything useful on https://github.com/FVELER/FVEL.
I am willing to adjust my score if the authors can help me locate their code or demos.


There are also several minor writing nitpicks:
* In the abstract, the FVELer dataset is not well-introduced. "To achieve this goal, we extract a large-scale FVELer". This sentence does not state that FVEler is a dataset. The authors may consider a rewrite of: "To achieve this goal, we present FVELer, a large-scale dataset of ..."
* On Line 95: "The field of automated theorem proving has developed formal languages such as first-order logic ...". This is a very dangerous claim! The authors should consider a different phrasing.
* Inconsistent capitalization when referring to the C language, e.g., Line 134, where "c" is used.

**Relation To Prior Work:**

The relation to prior work is sufficiently discussed.

**Summary And Contributions:**

This paper presents the FVEL environment for automated theorem proving with LLMs.
The FVEL environment takes as input C code, uses an LLM to derive specifications admissible for the Isabelle theorem-proving assistant, and then iteratively generates Isabelle proofs of the specification.
This tight integration of tooling is important for making formal verification more accessible to programmers and researchers.
The authors additionally present the FVELer dataset, which contains a large amount of C code, their associated specifications, and Isabelle proofs.

---

> ### Author Rebuttal · Authors · 2024-08-17
>
> ## Response to Reviewer BbdW
> ### 1. Release code and demo.
> > At present, my biggest concern is that I cannot find the code/demos for FVEL, and I would appreciate it if the authors could help me locate them. Because the FVEL framework is a central contribution, this work does not feel complete without the code. Unfortunately, I could not find anything useful on https://github.com/FVELER/FVEL. I am willing to adjust my score if the authors can help me locate their code or demos.
>
> Thank you for your concern.
> The code and related documentation of FVEL have been released:
>
> - **Datasheet:** It is already included in Appendix C.2 (Please see the Supplementary Material PDF).
> - **Code of extraction:** https://github.com/FVELER/FVELerExtraction
> - **l4v_FVEL:** [https://github.com/FVELER/l4v_FVEL](https://github.com/FVELER/l4v_FVEL)
> - **PISA_FVEL:** [https://github.com/FVELER/PISA_FVEL](https://github.com/FVELER/PISA_FVEL)
> - **Code of evaluation:** [https://github.com/FVELER/FVEL](https://github.com/FVELER/FVEL)
> - **Demonstrations and Demo:** Some demonstrations of FVEL have been shown on our [project page](https://fveler.github.io). We will develop a full demo web page, which supports interactive formal verification with LLMs online.
>
>
>
> ### Typos.
> > In the abstract, the FVELer dataset is not well-introduced. "To achieve this goal, we extract a large-scale FVELer". This sentence does not state that FVEler is a dataset. The authors may consider a rewrite of: "To achieve this goal, we present FVELer, a large-scale dataset of ..."
> > On Line 95: "The field of automated theorem proving has developed formal languages such as first-order logic ...". This is a very dangerous claim! The authors should consider a different phrasing.
> > Inconsistent capitalization when referring to the C language, e.g., Line 134, where "c" is used.
>
>
> Thanks for the detailed review. We have corrected the typo and inappropriate expression accordingly：
>
> Abstract: "we extract a large-scale FVELer ..." -> "we extract a large-scale dataset for automated formal verification named FVELer ..."
>
> Line 95: "the field of automated theorem proving (ATP) has developed formal languages such as first-order logic (FOL) and higher-order logic (HOL) to describe mathematical problems, theorems, and solution processes, allowing deductive reasoning to achieve the final answer or proof with rigorous stepwise validation." -> "The field of automated theorem proving (ATP) utilizes formal languages such as first-order logic (FOL) and higher-order logic (HOL) to describe mathematical problems, theorems, and solution processes, allowing rigorous step-by-step validation through deductive reasoning to achieve final answers or proofs."
>
> Line 134, Line 141, Line 152, Line 259: all lowercase "c" have been revised to uppercase "C".
>
> We also have done a thorough grammar check and revise our manuscript.

---

> > ### Comment · Reviewer_BbdW · 2024-08-26
> >
> > Thanks for the response and code release. I have increased my score.

---

### Official Review · Reviewer_XkcJ · 2024-07-25
**Review of submission 612**

**Rating:** 6
**Confidence:** 2
**Correctness:** Yes
**Clarity:** Yes

**Review:**

The motivation of the paper is sound and clear. The proposed environment has enough details and is clearly illustrated. Statistics of the environment is shown as well. The paper is well-written and easy to follow. Here are some comments from my side.

- Figure 3 shows the different distributions of dependency by theory and lemma, which are quite different. Why 3(a) has a long-tail pattern while 3(b) doesn't? What is the distribution of other current similar environments? Does the proposed one look like them? Similar questions to Figure 4.

- For the compared method, I wonder if the SMT-based solver should be tested as a baseline as it is still widely used in many fields. More ablation study should be conducted, e.g. different token numbers, prompts and hyperparameters etc.

- It is not clear the difference between SV-COMP and SV-COMP-47, why the performance on them is different in Table 2.

**Strengths:**

See Review part.

**Additional Feedback:**

See Review part.

**Documentation:**

Yes

**Limitations:**

In the supplementary material.

**Opportunities For Improvement:**

See Review part.

**Relation To Prior Work:**

Yes

**Summary And Contributions:**

The paper proposes an interactive formal verification environment with LLMs for automated theorem proving, by first converting input code into Isabelle and feeding it to LLMs. The FVEL environment and its resources (FVELer) are introduced in detail. Benchmark shows the results of fine-tuned LLMs on two datasets using the proposed environment.

---

> ### Author Rebuttal · Authors · 2024-08-17
>
> ## Response to Reviewer XkcJ
> ### 1. Different distributions of dependency by theory and lemma.
>
> > Figure 3 shows the different distributions of dependency by theory and lemma, which are quite different. Why 3(a) has a long-tail pattern while 3(b) doesn't? What is the distribution of other current similar environments? Does the proposed one look like them? Similar questions to Figure 4.
>
> Thank you for your careful reading and consideration. The distributions for dependency by theory and lemma are correlated and depend on the structure of the source l4v library.
> In Isabelle, a lemma is a small, reusable proof unit used to prove other theorems in a theory, while a theory is a structured collection of definitions, lemmas, theorems, and proofs that provides a framework for logic and functionality. Therefore, the theorems with larger depth in Figure 3(b) are concentrated in a few specific theory files, resulting in the long-tail effect in Figure 3(a).
> To our knowledge, most similar environments [1,2] have been predicated upon extensive data filtration processes, which are not close to the actual situation and lack analysis of the original data distribution.
>
> When counting proofsteps in Figure 4, a long-tail pattern is a common distribution. This is because simple theorems (usually provable within a dozen steps) account for the majority of the theorem libraries, while complex theorems (requiring hundreds or even thousands of steps to prove) account for only a small number. Similar distributions can be seen in theorem libraries of various formal systems, such as mathlib in Lean and HOL in Isabelle.
>
> [1] Zhang, Lichen, Shuai Lu, and Nan Duan. "Selene: Pioneering Automated Proof in Software Verification." arXiv preprint arXiv:2401.07663 (2024).
>
> [2] Chuyue Sun, Ying Sheng, Oded Padon, and Clark W. Barrett. 2023. Clover: Closed-loop verifiable code generation. CoRR, abs/2310.17807.
>
> ### 2. SMT-based solver baselines.
> > For the compared method, I wonder if the SMT-based solver should be tested as a baseline as it is still widely used in many fields.
>
> Thank you for your concern. The symbolic baselines we compared are both SMT-based solvers.
>
> Uautomizer [3] is part of the Ultimate framework, which is a collection of tools designed for software verification. The UAutomizer works by converting a program to an intermediate representation, and then checking the properties of the program using an SMT solver.
>
> The pipeline of ESBMC [4] (Efficient SMT-Based Context-Bounded Model Checker) is similar to Uautomizer. It implements advanced incremental BMC and k-inductive proof-rule algorithms based on SMT and constraint programming (CP) solvers.
>
> We will add a more detailed description of the baseline in the revised version.
>
> [3] Ultimate Automizer and CommuHash Normal Form - (Competition Contribution). Heizmann, Barth, Dietsch, Fichtner, Hoenicke, Klumpp, Naouar, Schindler, Schüssele and Podelski, TACAS (SV-COMP) 2023, https://doi.org/10.1007/978-3-319-89963-3_30
>
> [4] Sá Menezes, R., Aldughaim, M., Farias, B., Li, X., Manino, E., Shmarov, F., Song, K., Brauße, F., R. Gadelha, M., Tihanyi, N., Korovin, K., & Cordeiro, L. C. ESBMC: 7.4: Harnessing the Power of Intervals (Version 7.4) [Computer software]. https://github.com/esbmc/esbmc
>
>
> ### 3. Ablation Study.
> > More ablation study should be conducted, e.g. different token numbers, prompts and hyperparameters etc.
>
> Thank you for your suggestion. The conclusions reported in the paper are all under the setting of max-token=2048. We are running ablation experiments with different max-token settings on the SV-COMP-1000 dataset for Mistral-7B, and the results are shown in the following table:
>
>
> |      Max tokens     | 256  | 512  | 1024 | 2048 | 4096 | 8192 |
> | -----------------   | ---- | ---- | ---- | ---- | ---- | ---- |
> | # Verified Program  |  17  |  25  |  59  |  84  |  87  |  88  |
>
> The results show that the limited context length is not enough to represent theorems and their proofs, resulting in only 17 and 25 programs being verified when max-token=256 and 512, respectively. max-token=2048 is a reasonable setting that takes into account both performance and efficiency. A larger context length is unlikely to bring significant performance improvements.
>
> More experiments with different prompts and hyperparameters are still in progress, and we will report the results in the appendix of the revised version and update the error ranges in Table 2 and Table 3.
>
> ### 4. Difference between SV-comp-47 and SV-comp.
> > It is not clear the difference between SV-COMP and SV-COMP-47, why the performance on them is different in Table 2.
>
> Thank you for your question. Due to space limitation, we have moved the details of the dataset to Appendix E.1. We will emphasize this in the revised version.
> The SV-COMP-47 dataset is selected by Lemur [5]. Each data point contains one or more loops, and both the symbolic solvers failed whitin the same timeout setting (20 minutes). The SV-COMP dataset contains richer formal verification samples, not only for loop invariants, but also for array out of bounds, null pointer checking, etc. SV-COMP-1000 is a randomly selected sample of 1000 points from the entire dataset, which is the most essential difference between SV-COMP-47 and SV-COMP-1000.
>
> [5] Wu, Haoze, Clark Barrett, and Nina Narodytska. "Lemur: Integrating Large Language Models in Automated Program Verification." The Twelfth International Conference on Learning Representations.

---

> > ### Comment · Reviewer_XkcJ · 2024-08-30
> >
> > Thanks for the rebuttal and clarification. I have no further concerns, and I am leaning toward accepting the paper.

---

### Official Review · Reviewer_Do8m · 2024-07-25
**Important dataset for finetuning LLMs to formalize code**

**Rating:** 7
**Confidence:** 3
**Correctness:** Yes.
**Clarity:** Yes

**Review:**

This is an original proposal to use LLM's capabilities in generating code to increase its formalization capabilities. FVELER is a large dataset (Almost 30k lemmas), and this is an important contribution. The paper is clearly written, with insightful statistics and information about FVELER (e.g. FIgure 3).

Regarding negative aspects, some typos were observed (line 297: "In special, it is difficult to generate an accurate lemma specification under uncertain loop conditions" -> in particular). Of course, the fact that only C code can be formalized is a major limitation.

**Strengths:**

See section on review.

**Additional Feedback:**

I am not sure, could sel4 be replaced with something else to enable Python code generation.

**Documentation:**

Datasheets for Datasets is missing (https://arxiv.org/abs/1803.09010)

**Limitations:**

Only C code is considered. Given that there are many papers in the domain of math-AI that focus on language models solving math problems by generating Python code (e.g. an early paper being https://www.pnas.org/doi/abs/10.1073/pnas.2123433119), it would be particularly beneficial to formalize Python code.

Since this might be difficult, because sel4 is tied to C, at least a comprehensive discussion should be included why using Python was difficult, and also what it would entail to enable Python code formalization.

**Opportunities For Improvement:**

See section on review.

**Relation To Prior Work:**

Yes.

**Summary And Contributions:**

A dataset, FVELER, together with an environment, FVEL, is proposed to aid transforming generated LLM code into Isabelle formal language, which verifies what that code does.  Code2Inv and SV-COMP benchmarks are used to assess how a selection of LLMs perform after being finetuned on FVELER. Performance increases of 12%-17% were observed.

---

> ### Author Rebuttal · Authors · 2024-08-17
>
> ## Response to Reviewer Do8m
> ### 1. Genearation to other programming language.
> > Only C code is considered. Given that there are many papers in the domain of math-AI that focus on language models solving math problems by generating Python code (e.g. an early paper being https://www.pnas.org/doi/abs/10.1073/pnas.2123433119), it would be particularly beneficial to formalize Python code.
> > Since this might be difficult, because sel4 is tied to C, at least a comprehensive discussion should be included why using Python was difficult, and also what it would entail to enable Python code formalization.
>
> We are optimistic that FVEL has the potential to be generalized to the formal verification of other programming languages. A simple implementation is to extend a new preprocessing step in the pipeline to translate high-level programming languages such as Python and JAVA into C and apply it to the existing FVEL framework. Since the current LLM has demonstrated excellent performance in [code translation](https://paperswithcode.com/task/code-translation) tasks, we believe that such generalization can be achieved in the near future. The application including but not limited to: system and software verification, solving mathematical problems, graphics and interface verification, and other related fields.
>
> ### 2. Missing datasheets for datasets.
> > Datasheets for Datasets is missing.
>
> Thank you for your concern. The datasheet is included in Appendix C.2 (Please see the Supplementary Material PDF).
>
> ### Typos.
> > line 297: "In special, it is difficult to generate an accurate lemma specification under uncertain loop conditions" -> in particular
>
> Thanks for the detailed review. We have corrected the typo accordingly and we also did a thorough grammar check in our revised manuscript.

---

> > ### Comment · Reviewer_Do8m · 2024-09-03
> >
> > I thank the authors for following up on my suggestions in the rebuttal below. I believe using LLMs for C to Python code translation is promising but could introduce errors on its own.
> > I recommend the authors mention/link to the datasheets from the supplementary material in the main body, as this will make it easier for the reader to discover the datasheets.
> >
> > All in all, I will keep my score. I believe this is a valuable resource paper for researchers working on AI for Math and LLMs.

---

### Official Review · Reviewer_9xRj · 2024-07-25
**Interactive Isabelle FV environment + dataset for better code verification**

**Rating:** 7
**Confidence:** 4
**Correctness:** Claims in the paper seem correct.

**Review:**

I have broken down my review into the next few specific sections. I like how this work is very complete and presents large-scale open resources for the community, as I discuss in the "Strengths" section. I leave my main questions under the "Opportunities For Improvement" sections, which I hope the authors can clarify. There are also some minor questions in the "Clarity" section to be corrected.

**Strengths:**

This work contributes large-scale valuable open resources to the theorem proving community especially for Isabelle. The work is solid and complete -- reporting from an environment to a dataset to finetuning results. It is also very relevant to the broader AI community, since theorem proving is one of the most difficult open problems in AI. I list some questions under the "Opportunities For Improvement" section and some minor suggestions under "Clarity".

**Additional Feedback:**

None. All feedback is already listed above.

**Clarity:**

The paper is generally clearly written and well structured. Its organization is pretty complex though, due to the many components it elaborates on. Some minor mistakes discovered during reading:

1. In Sec. 2, "proof assistances" -> "proof assistants".

2. In Sec. 3, "Then we initial the Isabelle state ... " -> "Then we initialize the Isabelle state ...".

3. The authors keep saying "program languages" whereas the exact terminology should be "programming languages".

**Documentation:**

Yes, the documentation seems clear.

**Ethics:**

There seems to be no ethical concern.

**Limitations:**

Limitations have been comprehensively discussed in Appendix A. There seems to be no direct societal impact with this work.

**Opportunities For Improvement:**

As the authors mentioned in the supplementary material, I would be very interested to see how FVEL & FVELer can be generalized to support more programming languages. I am curious about any anticipated challenges of using other formal verification languages, such as Coq. I know Coq has good library support for formal verification too. Is there anything specific preventing the authors' reported methods to generalize to Coq?

Also, while this paper studies C, I would be interested to know if there is any known difficulty expanding to other languages? E.g. python, java, etc. have all been widely used languages (esp. python), and some languages share similar origins as C/C++. What, if any, is stopping the methods to apply to those other languages?

In FVEL, there are multiple steps in the pipeline, including converting C/C++ code to Isabelle formal code (code formalization/translation), generate a lemma, prove the lemma (which may need multiple steps, possibly), etc. Each of the steps sound like very different yet all non-trivial tasks for LLMs. I'm curious which of the tasks are the most difficult as observed during experiments, and which ones are trivial. For the hardest ones, is there any attempt to better solve them yet?

**Relation To Prior Work:**

The discussion of prior work is clear.

**Summary And Contributions:**

This paper introduces FVEL, an interactive Isabelle formal verification (FV) environment, and FVELer, a large-scale dataset extracted therefrom for finetuning and/or evaluation. The authors conclude through experiments that FVEL with LLMs finetuned on FVELer achieves performance improvement on code verification task.

---

> ### Author Rebuttal · Authors · 2024-08-17
>
> ## Response to Reviewer 9xRj
> ### 1. Generalization to Coq or other formal system.
> > As the authors mentioned in the supplementary material, I would be very interested to see how FVEL & FVELer can be generalized to support more programming languages. I am curious about any anticipated challenges of using other formal verification languages, such as Coq. I know Coq has good library support for formal verification too. Is there anything specific preventing the authors' reported methods to generalize to Coq?
>
> Yes. There are tools for applying Coq to formal verification, such as [CompCert](https://github.com/AbsInt/CompCert). As for complex system, CertiKOS [1] is a practical concurrent OS kernel and verified its (contextual) functional correctness in Coq.
> However, the amount of data generated by existing Coq-based formal verification work is limited, and CertiKOS only contains 6,500 lines of C and x86 assembly code. Extensive translation of C into other formal languages such as Coq requires a more mature toolchain.
> We will initiate our efforts with FVEL as a foundational reference, and collaborate with the community in our ongoing endeavors to foster an expansion of verification studies across a spectrum of diverse formal systems.
>
> [1] Ronghui Gu, Zhong Shao, Hao Chen, Xiongnan Wu, Jieung Kim, Vilhelm Sjöberg, and David Costanzo. 2016. CertiKOS: an extensible architecture for building certified concurrent OS kernels. In Proceedings of the 12th USENIX conference on Operating Systems Design and Implementation (OSDI'16). USENIX Association, USA, 653–669.
>
> ### 2. Generalization to Python or other programming language.
> > Also, while this paper studies C, I would be interested to know if there is any known difficulty expanding to other languages? E.g. python, java. have all been widely used languages (esp. python), and some languages share similar origins as C/C++. What, if any, is stopping the methods to apply to those other languages?
>
> We are optimistic that FVEL has the potential to be generalized to the formal verification of other programming languages. A simple implementation is to extend a new preprocessing step in the pipeline to translate high-level programming languages such as Python and JAVA into C and apply it to the existing FVEL framework. Since the current LLM has demonstrated excellent performance in [code translation](https://paperswithcode.com/task/code-translation) tasks, we believe that such generalization can be achieved in the near future. The application including but not limited to: system and software verification, solving mathematical problems, graphics and interface verification, and other related fields.
>
> ### 3. The hardest task for LLMs in the pipeline.
> > In FVEL, there are multiple steps in the pipeline, including converting C/C++ code to Isabelle formal code (code formalization/translation), generate a lemma, and prove the lemma (which may need multiple steps, possibly). Each of the steps sound like very different yet all non-trivial tasks for LLMs. I'm curious which of the tasks are the most difficult as observed during experiments, and which ones are trivial. For the hardest ones, is there any attempt to better solve them yet?
>
> Thank you for your question.
> We believe that pre-trained/fine-tuned language models can understand the semantics of code/formal theorems well and prove theorems in a one-pass/step-by-step manner with search methods. For LLMs, the main difficulty lies in generating lemma specification corresponding to the semantics according to different requirements. Besides, there is no rigorous metric to check the alignment of such semantics yet.
>
>
> ### Typos.
> > In Sec. 2, "proof assistances" -> "proof assistants".
> > In Sec. 3, "Then we initial the Isabelle state ... " -> "Then we initialize the Isabelle state ...".
> > The authors keep saying "program languages" whereas the exact terminology should be "programming languages".
>
> Thanks for the detailed review. We have corrected the typos accordingly:
> Line 99: "proof assistances" -> "proof assistants"
> Line 143: "Then we initial the Isabelle state ... " -> "Then we initialize the Isabelle state ..."
> Line 50, Line 128: "program languages" -> "programming languages"
>
> We also have done a thorough grammar check and revised our manuscript.

---

> ### Comment · Reviewer_9xRj · 2024-08-17
> **Reviewer Response to Author Rebuttal**
>
> Thank the authors for the detailed responses to each and every of my questions.
>
> ### Re [1]:
>
> This makes sense. I think Coq has been important in many software/hardware verification works, and it would be great if the authors' work can be generalized especially to Coq. Yet I would not ask the authors for any additional efforts or more experiments on this, as I feel this can be, exciting though, kind of orthogonal to this work, plus that a different formal language is likely to require some different handling. I do look forward to the authors' continued great work on this generalization.
>
> As such, I would just follow up with a quick conceptual question: I agree with the authors that "extensive translation of C into other formal languages such as Coq requires a more mature toolchain." Would you elaborate on this and explain in which specific ways the toolchain can be made more "mature" in order to accomplish this generalization? The authors mentioned collaborating with the community as well -- and I completely agree with this, especially with Coq's great community. Are there any specific ideas on how the authors would like to initiate the collaboration, such as what the authors would like to gain support on?
>
> ### Re [2]:
>
> I find this line of generalization (to different programming languages) even more central than the generalization to other formal languages. I like the authors' proposal of the "simple implementation", and I quite like the idea. I would be very interested to see some early results on that. Just to be clear -- I am not expecting any comprehensive studies or completely another work. Some initial experiments of what the authors think are simple to implement would do the job, as a proof of concept that the proposed framework does have an ability to generalize along this direction.
>
> ### Re [3]:
>
> Intuitively I would agree with the discussion too. Thanks for answering.
>
> ### Re [Typos]:
>
> Thanks for fixing them. I like how the authors make an effort to improve the quality of the presentation.

---

> ### Author Response · Authors · 2024-08-17
> **Response to Reviewer 9xRj**
>
> ### 1. C to Coq.
>
> Our ideal C to Coq conversion tool can automatically and provably abstract low-level C language semantics into higher-level representations, similar to C-parser and AutoCorres, to facilitate manual reasoning and simplify verify process. Of course, if we can measure the semantic alignment before and after the conversion, then we may be able to use LLM to achieve this process. Therefore, the direction we hope to cooperate with the Coq community may be:
>
> - Tool-chain development. This process may include generating abstract syntax trees, extracting function call relationships, translating into Coq theorems, providing proofs, ..., which requires various knowledge backgrounds.
>
> - Data annotation platform. It may be a promising direction to construct C to Coq data through annotations by community members, and train a LLM-based translator with open-source data.
>
> ### 2. Python / Java verification.
> Thank you for your inspiring comment. We are currently constructing experiments with this idea. Ready to update you on the latest results!

---

> > ### Comment · Reviewer_9xRj · 2024-08-17
> > **Look forward to seeing the results**
> >
> > Thank the authors for the further response.
> >
> > ### C to Coq
> >
> > The plans sound promising, thanks for sharing your thoughts.
> >
> > ### Python / Java verification
> >
> > Sounds great -- look forward to seeing the results.

---

> > > ### Author Response · Authors · 2024-08-27
> > > **Attempt to verify Python code.**
> > >
> > > Thank you for recognizing our ideas. We have some preliminary conclusions. The details of our experiment are as follows:
> > >
> > > 1. Python / Java to C data source. We investigated most of the open-source code translation datasets. However, few datasets provide aligned Python / Java to C data samples. The C-code in most datasets is written in C++, while the C-parser tool only supports the C99 standard. Considering that it is difficult to use automated tools or rules to normalize these C++ codes to C99, we do not use open-source datasets such as HumanEval-X[1], CodeNet[2], and so on. Instead, we collected some algorithmic solutions implemented in different programming languages from the Online Judge platform ([Leetcode](https://leetcode.com/) and [POJ](http://poj.org/)) and manually checked their semantic equivalence. Finally, we obtained a small test dataset of 93 samples, each data point contains a Python solution as well as a C solution. One of the data points is shown below:
> > >
> > >
> > > C code:
> > > ```c
> > > int removeDuplicates(int* nums) {
> > >     int numsSize = sizeof(nums) / sizeof(nums[0]);
> > >     int j = 1;
> > >     for (int i = 1; i < numsSize; i++) {
> > >         if (nums[i] != nums[i - 1]) {
> > >             nums[j] = nums[i];
> > >             j++;
> > >         }
> > >     }
> > >     return j;
> > > }
> > > ```
> > >
> > > Python code:
> > > ```python
> > > def removeDuplicates(nums: List[int]) -> int:
> > >     j = 1
> > >     for i in range(1, len(nums)):
> > >         if nums[i] != nums[i - 1]:
> > >             nums[j] = nums[i]
> > >             j += 1
> > >     return j
> > > ```
> > >
> > >
> > > 2. Code translation. We use GPT-4 to translate the collected Python code into C. We ensure that GPT-4 generates C99-standard code by prompt engineering, and evaluate the quality of the generated code by using the success rate of C-parser parsing as well as the CodeBLEU[3] value with respect to ground-truth. GPT-4o gets 84.94% (79/93) passrate and 81.32 CodeBLEU score:
> > >
> > > |                   |  Pass rate  |   CodeBLEU  |
> > > | ----------------- |:-----------:|:-----------:|
> > > |       GPT-4o      |    84.94%   |    81.32    |
> > >
> > >
> > > This shows that advanced LLM can handle code translation tasks. The prompt is shown below.
> > >
> > > > ###Prompt: You are an excellent programmer. Below is a piece of Python code that you are asked to translate into C99 standard code. Be careful not to use C++ syntax, functions, and features. Avoid GOTO statement, Union, fall-through cases in switch statements.
> > > >
> > > > <<Python Code>>
> > >
> > > 3. Verification result. We then apply the translated C code as input to our framework and verify it using our baseline models. The verification result is shown in the following table:
> > >
> > >
> > > |                 | # Verified  |
> > > | --------------- |:-----------:|
> > > | Llama-3-8B      |    35/93    |
> > > | Llama-3-8B-FVEL |    42/93    |
> > > | Mistral-7B      |    38/93    |
> > > | Mistral-7B-FVEL |    43/93    |
> > >
> > >
> > > Conclusion: Through the above experiments, we believe that the FVEL framework has the potential to be migrated to the verification of code in other programming languages. Pretrained LLM has already demonstrated great power in the verification of simple Python programs that don't involve multiple layers of import; with fine-tuning, the performance improves even further.
> > >
> > > However, the current support for only C99 standard code remains one of the biggest limitations. Our subsequent work hopes to promote the generalization of FVEL in collaboration with the community (mentioned in Q1).
> > >
> > > The above datasets, experiments and conclusions will be added to the revised version of the paper. If you have further questions and comments, please feel free to contact us.
> > >
> > >
> > > [1] Zheng, Qinkai, et al. "Codegeex: A pre-trained model for code generation with multilingual benchmarking on humaneval-x." Proceedings of the 29th ACM SIGKDD Conference on Knowledge Discovery and Data Mining. 2023.
> > >
> > > [2] Hwang, Seon-Jin, et al. "CodeNet: Code-targeted convolutional neural network architecture for smart contract vulnerability detection." IEEE Access 10 (2022): 32595-32607.
> > >
> > > [3] Ren, Shuo, et al. "Codebleu: a method for automatic evaluation of code synthesis." arXiv preprint arXiv:2009.10297 (2020).

---

> > > > ### Comment · Reviewer_9xRj · 2024-08-27
> > > > **Results look good; raising score**
> > > >
> > > > Thank the authors for the additional experiment. The POC of using a LLM for code translation and then applying FVEL for verification sounds reasonable, and the results make sense. If the authors later have extra space in their camera ready version (if accepted), I would encourage the authors to add these results to support the argument of FVEL's generality.
> > > >
> > > > I would like to thank the authors again for the thorough rebuttal. It has been very effective. In my initial review, I gave a score of 6 with confidence 3. During the author-reviewer discussion so far, the authors have clarified all the points I raised about possible miscommunication / presentation issues, and promised to update these points in the cam-ready version if accepted. These efforts have cleared most of my previous uncertainty from their presentation, making me more confident in my review. So now, I would classify my confidence score as 4.
> > > >
> > > > Now with the additional results supporting this methods' generalizability, I think this paper not only helps with the specific application the authors report but also opens a large space for downstream uses, so I am raising my score to recommend acceptance (7).
> > > >
> > > > Thank the authors for the great work. Look forward to your future investigations in related areas.

---

### Official Review · Reviewer_WawV · 2024-07-26
**The review of FVEL.**

**Rating:** 6
**Confidence:** 4
**Correctness:** Yes.
**Clarity:** Yes.

**Review:**

Strength:

1. The authors have made significant contributions by not only creating a large-scale formal verification dataset but also designing an interactive environment that enhances the usability of the dataset.

2. The FVEL dataset has proven its effectiveness as all the LLMs showed improved performance post-finetuning.

3. This paper is well-organized and easy to understand.



Weakness:

1. After finetuning Llama3-8B, the solved problems in SV-Comp improve from 69 to 81. However, the question remains whether the effort of finetuning is worthwhile for a marginal increase of only 12 problems.

2. The authors claim that symbolic solvers are time-consuming, despite their ability to solve a higher number of problems in SV-COMP compared to LLM-based solvers. However, there is no runtime comparison in Table 2 to support this claim.

**Strengths:**

See review.

**Additional Feedback:**

See review.

**Documentation:**

Yes.

**Ethics:**

No.

**Limitations:**

See review.

**Opportunities For Improvement:**

See review.

**Relation To Prior Work:**

Yes.

**Summary And Contributions:**

This paper proposes FVEL, an interactive Formal Verification Environment with LLMs, and extracts a large-scale dataset, FVELER, which includes in-depth code dependencies and verification processes from Isabelle. Experimental results show that the FVELER fine-tuned Llama3-7b and Mistral-7B can solve more problems in verification benchmarks.

---

> ### Author Rebuttal · Authors · 2024-08-17
>
> ## Response to Reviewer WawV
> ### 1. Whether the effort of finetuning is worthwhile.
> > After finetuning Llama3-8B, the solved problems in SV-Comp improve from 69 to 81. However, the question remains whether the effort of finetuning is worthwhile for a marginal increase of only 12 problems.
>
> We believe that the reason for similar problems is that SV-COMP is somewhat out-of-distribution as a downstream dataset. Although both datasets are derived from the validation of complex systems, the programs in SV-COMP are relatively independent, and the difficulty in verification lies in the code length and complex logic; while the main difficulty of the theorems in the FVELer dataset lies in the dependent relationship. Therefore, SV-COMP is used as an evaluation of the generalization ability of the model after fine-tuning, and the absolute improvement of solving 12 problems is quite significant.
>
> In addition, the results of fine-tuning on the in-domain dataset (our FVEL-test and FVEL-test-hard, see Appendix D) show that fine-tuning is worthwhile (26 -> 74 improvement for Mistral, 39 -> 88 for Llama-3 on test set, and 19 -> 49 for Mistral, 27 -> 64 for Llama-3 on test-hard set).
>
> ### 2. Evidance of time-consuming symbolic solver.
> > The authors claim that symbolic solvers are time-consuming, despite their ability to solve a higher number of problems in SV-COMP compared to LLM-based solvers. However, there is no runtime comparison in Table 2 to support this claim.
>
> Thank you for pointing this out. In our experiment, the solvers need to solve each problem within a same timeout setting (10 minutes and 20 minutes for Code2Inv and SV-COMP, respectively). Since it is unfair to directly compare the running time of methods running on the CPU and methods running on the GPU, we gave symbolic solvers a maximum timeout of two hours on the SV-COMP dataset and found that, disappointingly, symbolic solvers often got stuck when verifying complex programs and did not solve more problems. Since it is difficult to solve this issue or develop highly parallel symbolic solvers, we claim that "symbolic SMT solvers are time-consuming and uneconomical when there is a large amount of code to be verified".

---

> > ### Comment · Reviewer_WawV · 2024-08-26
> >
> > Thank the authors for the efforts in the rebuttal. I will keep my original score.

---

### Author Rebuttal · Authors · 2024-08-17

## General Response

We thank all the reviewers for taking the time to assess our submission. We are grateful for the insightful comments and are thankful for the positive recognition of the significant motivation and contributions (Reviewers WawV, 9xRj, Do8m, XkcJ, BbdW), novelty (Reviewers Do8m), solidness (Reviewers 9xRj), effectiveness (Reviewers WawV) and clear presentation (Reviewers WawV, Do8m, XkcJ) of our benchmark.

For reviewers' concerns, in the following we provide a unified response to some common questions:

### 1. Release of code and demo (Reviewers BbdW, Do8m).
The code and related documentation of FVEL have been released:

- **Datasheet:** It is already included in Appendix C.2 (Please see the Supplementary Material PDF).
- **Code of data extraction:** https://github.com/FVELER/FVELerExtraction
- **l4v_FVEL:** [https://github.com/FVELER/l4v_FVEL](https://github.com/FVELER/l4v_FVEL)
- **PISA_FVEL:** [https://github.com/FVELER/PISA_FVEL](https://github.com/FVELER/PISA_FVEL)
- **Code of evaluation:** [https://github.com/FVELER/FVEL](https://github.com/FVELER/FVEL)
- **Demonstrations and Demo:** Some demonstrations of FVEL have been shown on our project page (https://fveler.github.io) at **Dataset Examples**. We will develop a full demo web page, which supports interactive formal verification with LLMs online.


### 2. Generalization of FVEL (Reviewers 9xRj, Do8m).
We thank the reviewers for their affirmation of our work and their expectation for further generalization. Currently, generalizing the FVEL framework to other formal systems requires richer data sources and more mature tool chains. We will continue to engage with the community and promote more formal verification work based on different formal systems in the future.

We are optimistic that FVEL has the potential to be generalized to the formal verification of other programming languages. A simple implementation is to extend a new preprocessing step in the pipeline to translate high-level programming languages such as Python and JAVA into C and apply it to the existing FVEL framework. Since the current LLM has demonstrated excellent performance in [code translation](https://paperswithcode.com/task/code-translation) tasks, we believe that such generalization can be achieved in the near future. The application including but not limited to: system and software verification, solving mathematical problems, graphics and interface verification, and other related fields.



Detailed rebuttal please check the individual responses below. We have made considerable efforts to respond to each reviewer. We hope our response addresses the reviewers' concerns.


Sincerely,

Authors of paper #612

---

### Decision · Program_Chairs · 2024-09-26

**Decision:**

Accept (Poster)

**Comment:**

We thank the authors for their submission and constructive engagement throughout the rebuttal phase.

The reviewers and AC were enthusiastic about several aspects of this work: the paper covers a timely and interesting question. We found their contributions to be practical and usable, but also providing sound technical grounding. The reviewers highlighted both the methodological strengths of the paper as well as the dataset, which is an interesting contribution in itself. Overall, we believe that this work will be of interest to the broad range of researchers and can inspire future work.

Though we support that the work be accepted for publication, we also strongly encourage the authors to address some of the key concerns that the reviewers brought up. These include:
(1) concerns around generalizability to other programming languages. We believe that it is important to include even preliminary results as a proof of concept and an extensive discussion for this iteration of the work to increase confidence.
(2) The reviewers pointed out several lower-level questions about the evaluation and presentation: e.g., Figure 3, clarifying difference btween SV-COMP and SV-COMP-47, more ablation studies, and so on. These additional evaluations and clarifications will increase confidence in the contributions.
(3) The reviewers and AC agree that the dataset itself is an interesting submission, but in how it is presented in the paper so far, it is easy to overlook that until far into the paper. It would be great for the authors to highlight this early in the work.
(4) while the reviewers appreciate the end-to-end nature of this work, the presentation at times confounds what tasks in the pipeline create bottlenecks and what parts may be trivial. A discussion and clarification around this would strengthen the work and allow the reader to hone in on interesting future questions.
(5) There were several points of confusion for the reviewers, several of which the authors clarify during the rebuttal phase, as well as various typos, wording suggestions, and other such small issues that distract from an otherwise strong submission. We encourage the authors to incorporate these in the camera-ready version of the work.

Overall, we appreciate the authors' contributions and look forward to reading the final version of the work / seeing the presentation.